# RS-MoE: Collaborative Compression for Mixture-of-Experts LLMs based on Low-Rank and Sparse Approximation

## Abstract

Mixture-of-Experts (MoE) based Large Language Models (LLMs), despite their computational efficiency, face significant storage and memory challenges, which hinder their deployment on edge devices. However, existing methods primarily focus on compressing at the expert level, resulting in the loss of specialized knowledge. To address these challenges, we propose a novel framework termed RS-MoE, which compresses MoE models by collaboratively decomposing the weights of each expert into low-rank and sparse components. Through a preliminary investigation of the relationship between activations and weights, we identified two key observations: (i) a small fraction of weight dimensions, identifiable by high activation peaks, are critical and can be treated as a sparse component, and (ii) the remaining weights, after removing these high-importance dimensions, exhibit an inherent low-rank structure. Building on this, we developed a comprehensive importance score based on activation peaks to apply a tailored policy: high-importance dimensions are sparsely preserved, while the remaining dimensions are approximated using a low-rank representation. Additionally, ridge regression and mutual information techniques are incorporated to further minimize errors. We performed a comprehensive evaluation of RS-MoE on several MoE LLMs, including DeepSeekMoE-16B-Base, Mixtral-8x7B, and Qwen3-30B-A3B. The results demonstrate that our approach consistently outperforms existing monolithic sparse or low-rank methods across a variety of downstream tasks, highlighting its superior effectiveness and generalizability.

## 1 Introduction

Large Language Models (LLMs) based on the Mixture-of-Experts (MoE) architecture (Cai et al., 2025) offer an innovative approach to tackling issues associated with scaling models through sparse activation (Kaplan et al., 2020), while maintaining comparable computational efficiency. Several representative models, including DeepSeek-V3 (DeepSeek-AI et al., 2025), Mixtral-MoE (Jiang et al., 2024b), and Qwen3-30B-A3B (Yang et al., 2025) have achieved outstanding performance in translation, code generation, and question answering tasks, indicating the effectiveness of MoE LLMs. However, the benefits of MoE's computational efficiency can be offset by challenges such as static storage overhead and memory access latency. It is becoming increasingly commonplace in resource-constrained devices like edge devices (Zhong et al., 2025b), revealing the necessity for MoE compression.

Several studies have focused on compression techniques to address the challenges mentioned above, which can be mainly classified into two kinds: expert pruning and expert merging. Firstly, expert pruning primarily achieves compression via removing redundant or low-importance experts from the network. Methods such as MoE-I[2] (Yang et al., 2024), NAEE (Lu et al., 2024), and MoE-Pruner (Xie et al., 2024) use different pruning strategies to assess the importance of each expert and perform pruning. However, expert merging may result in significant performance degradation due to the loss of specialized knowledge, especially at a high compression ratio. Secondly, expert merging identifies similarities among experts to combine those that are highly similar. Techniques like MC-SMoE (Li et al., 2024), HC-SMoE (Chen et al., 2025), and Sub-MoE (Li et al., 2025) merge expert weights via weighting or clustering. Although expert merging preserves the model's functional integrity, it might

dilute specialized expertise, leading to a degradation of overall performance. These drawbacks of existing methods prompt us to consider: **Is there a novel compression paradigm that can not only preserve the diversity of experts but also avoid damaging the integrity of each expert?**

The recent study leverages the sparsity of the input activations and the low-rank approximation of the weights to achieve low-loss inference acceleration (Zhang et al., 2025). Inspired by this approach, we aim to determine whether the weight matrix can be decomposed into its sparse and low-rank components to capture the essential information contained in the weights. However, identifying an appropriate basis for decomposing the weights presents an additional challenge. Another research suggests that only a small number of experts significantly influence the performance of the MoE (Su et al., 2025). Moreover, identifying these influential experts relies more on the intensity of their activation peaks rather than the magnitude of their weights or the frequency of their activations. Building upon this, we can access the importance of each dimension of the expert and decompose the weight into sparse and low-rank components according to the activation peaks. In order to validate this assumption, we examine the expert's distributions of activation peaks and the singular spectrum of the weight that is whitened by the input of the expert, as illustrated in Figure 1. It becomes evident that the distributions of activation peaks can be divided into three categories: high importance, medium importance, and low importance. Otherwise, it can be observed that the energy of the weight matrix, with its high-importance components zeroed out, is concentrated in a few singular vectors, revealing a low-rank structure. To summarize, we can show that experts' weights can be approximated using sparsity and low-rank decomposition without significantly degrading information.

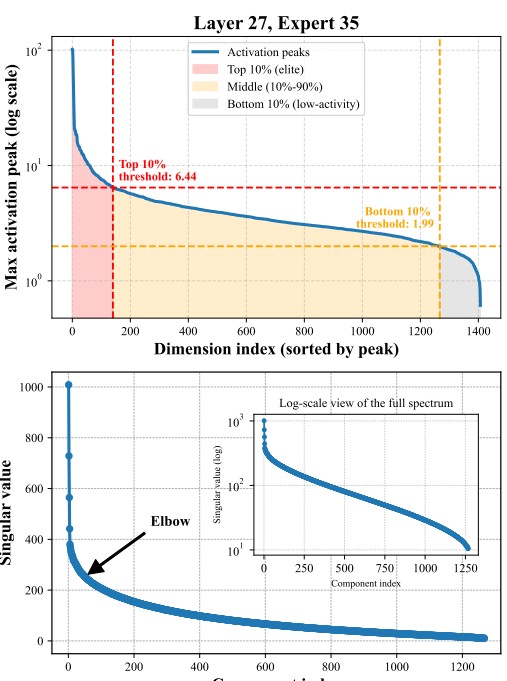

Figure 1: Analysis of Activation Sparsity and Low-Rank Structure. (Top) The sorted activation peaks reveal a sparse pattern where only a few dimensions are highly active. (Bottom) After pruning the top 10% of rows identified by these activations, the remaining weight matrix exhibits a strong low-rank property after activation-aware SVD. More detail can be found in Appendix A.4.

Building upon this, We propose a novel collaborative compression strategy for MoE LLMs, termed **RS-MoE**, which is developed based on the coupling relationships among internal weights within experts. Specifically, by analyzing the interactions between activation vectors and weight vectors, we establish a direct dimensional mapping relationship. This mapping couples the corresponding dimensions of the three weight matrices within experts through intermediate activations. By applying a unified compression strategy to these coupled dimensions, we effectively mitigate errors arising from spatial misalignment. The **main contributions** of this paper are summarized as follows:

- To preserve the functional integrity within MoE experts, we propose a novel collaborative framework that couples the corresponding dimensions of the three expert weights into a collaborative unit and performs the same compression strategy.

- To distinguish sparse components from low-rank components, we design a comprehensive importance score to evaluate the importance of each weight dimension. Dimensions with high importance are regarded as sparse components, while those with medium and low importance are classified as low-rank components using activation-aware SVD. Subsequently, ridge regression is applied to learn a shared base weight that compensates for the overall reconstruction error.

- To effectively allocate the sparsity ratio across layers, we estimate parameter redundancy in each layer based on the mutual information of activations between adjacent layers, thereby implementing a layer-aware compression strategy.

- To comprehensively evaluate the effectiveness of RS-MoE, we have conducted extensive experiments on three representative MoE-based LLMs: DeepSeekMoE-16B-Base, Mixtral-8×7B, and Qwen3-30B-A3B. The proposed RS-MoE demonstrates state-of-the-art performance across a wide range of downstream tasks and sparsity rates. Notably, RS-MoE exhibits significant advantages, particularly under high sparsity rates.

## 2 RELATED WORK

### 2.1 LARGE LANGUAGE MODELS COMPRESSION

LLMs require a tremendous amount of computational resources because of their parameter scale, which limits their use on devices with restricted resources. In related research, many approaches have been proposed to address LLMs' high storage and computational demands. One common method is model quantization, which reduces storage demands by converting model weights into lower-bit representations (Dettmers et al., 2022; Frantar et al., 2022; Lin et al., 2024). Another approach is model pruning and sparsification, which removes redundant parameters while minimizing performance degradation, thus refining the model structure (Ma et al., 2023; Frantar & Alistarh, 2023; Liu et al., 2023). Knowledge distillation is also a widely used technique for compressing LLMs. It involves training a smaller "student" model to replicate the behavior of a larger "teacher" model, enabling effective knowledge transfer (Acharya et al., 2024; Gu et al., 2025b). Otherwise, low-rank decomposition is often used to reduce model complexity by factorizing weight matrices and retaining their principal energy components (Wang et al., 2025c;a;b).

### 2.2 LOW-RANK AND SPARSE APPROXIMATION

Several studies have validated the effectiveness of low-rank approximation and sparsification for model compression. First, the weights of LLMs are always over-parameterized, which means that their intrinsic rank is usually lower than the original dimensions (Hu et al., 2022). Methods such as SVD or projection can extract the principal components of weight matrices, allowing for the approximation of the matrix using a low-rank representation (Yu & Wu, 2023; Wang et al., 2025c). In addition, sparsification methods identify and remove redundant weights to accelerate inference and reduce computational costs (Sun et al., 2024), which are often based on activation strength. Nevertheless, both techniques are limited: low-rank approximation can struggle to represent high-rank or multi-modally distributed weights. At the same time, structured sparsification cannot maintain model performance at high compression rates. Recently, some studies have explored the combination of low-rank and sparse representations to reduce the number of parameters while preserving critical structural information (Li et al., 2023; Huang et al., 2025a;b). Nevertheless, LoSparse requires expensive iterative retraining due to its additive decomposition. While SoLA is training-free, it relies on simple activation norms that overlook the specific activation peaks critical for MoE experts. Furthermore, neither method addresses the structural coupling in SwiGLU-based experts. In contrast, RS-MoE introduces a collaborative decomposition that preserves this functional alignment and retains expert specialization without retraining.

## 3 METHODOLOGY

### 3.1 PRELIMINARIES

In this paper, we treat the compression of MoE LLMs as a layer-wise reconstruction problem, aiming to minimize the adverse effects on the compressed output of each layer. Consider a typical MoE architecture, where each block contains three types of linear layers: attention weights, gating weights, and expert weights. Notably, the expert weights typically constitute over 90% of the entire model's parameters. Consequently, we only compress the expert weights to meet the overall sparsity ratio in the experiment.

An expert is generally consist of three matrices: $\mathbf{W}_{up}, \mathbf{W}_{gate} \in \mathbb{R}^{m \times n}$ and $\mathbf{W}_{down} \in \mathbb{R}^{n \times m}$, where $n$ and $m$ respectively denote the dimension of model hidden and intermediate activations. The computational process of the expert can be expressed as: $Y = gH\mathbf{W}_{down}^{\top}$, where $H = X\mathbf{W}_{up}^{\top} \odot \sigma(X\mathbf{W}_{gate}^{\top})$ and $g$ represents the routing score assigned to the expert, as resolved by the gating network.

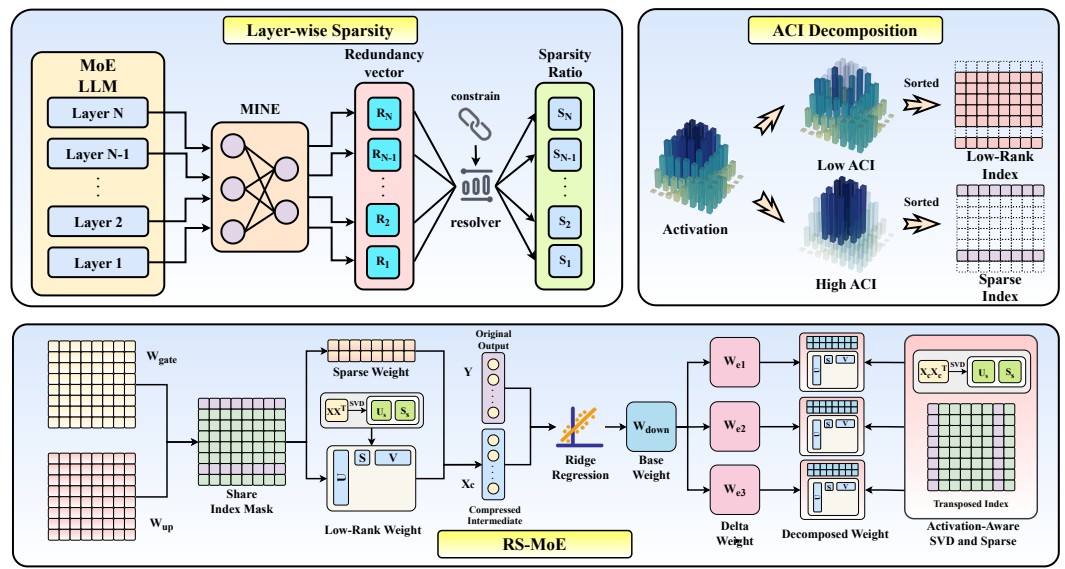

Figure 2: Overview of the RS-MoE. The process consists of three main steps: (1) Estimate mutual information to obtain layer-wise sparsity ratios. (2) Evaluate importance using Anomalous Contribution Integration, decompose weights into sparse and low-rank components. (3) Implement a tailored strategy to compress weights collaboratively.

## 3.2 THE RS-MOE FRAMEWORK: COLLABORATIVE DECOMPOSITION

The traditional MoE compression method often regards each expert as an independent entity when pruning or merging, which tends to destroy the complex relation inside the expert and cause severe knowledge loss. Therefore, as shown in Figure 2, we introduce a novel framework named RS-MoE, which treats the weights of experts as a coupled entity. This coupling is particularly evident in the SwiGLU architecture, which is widely adopted in MoE LLMs. As mentioned in the former section, the expert's three weight matrices are directly mathematically linked via the intermediate activation $H$. Specifically, given an input vector $x \in \mathbb{R}^{1 \times n}$, the $j$-th component of the intermediate activation $h \in \mathbb{R}^{1 \times m}$ is exclusively determined by the $j$-th rows of the weight matrices $\mathbf{W}_{up}, \mathbf{W}_{gate} \in \mathbb{R}^{m \times n}$:

$$h_j = \sigma(x \cdot \mathbf{W}_{gate,j,:}) \odot (x \cdot \mathbf{W}_{up,j,:})$$

Subsequently, this activation vector $H$ is projected by $\mathbf{W}_{down}$ to produce the output. We can regard the output as a linear combination of the columns of $\mathbf{W}_{down}$ with the elements of H serving as the coefficients.

$$Y = g \sum_{j=1}^{m} h_j \mathbf{W}_{down,:j}$$

Therefore, we can unfold the expert computation into a single summation over its intermediate dimensions:

$$Y = g \sum_{j=1}^{m} (\sigma(x \cdot \mathbf{W}_{gate,j,:}) \odot (x \cdot \mathbf{W}_{up,j,:}))\mathbf{W}_{down,:j}$$

Apparently, this equation indicates that the column $j$ of $\mathbf{W}_{down}$ directly connects to the row $j$ of $\mathbf{W}_{up}$ and $\mathbf{W}_{gate}$, establishing a collaborative unit. Building upon this, we further propose a tailored decomposition for fine-grained collaborative units based on their importance. We decompose each

weight into two components: (i) A sparse component, which preserves high-importance weight to ensure the integrity of the vital knowledge. (ii) A low-rank component, whose energy is concentrated in a few singular values, maintains the expressive capability with a small number of parameters. Our collaborative decomposition strategy preserves the essential part of expert knowledge, preventing information loss or blending.

## 3.3 ANOMALOUS CONTRIBUTION INTEGRATION

To accurately evaluate the importance of each collaborative unit and thereby differentiate between sparse and low-rank components, inspired by Su et al. (2025), we analyze the distribution of the intermediate activations. As shown in Figure 1, the majority of activations are generally relatively low, while only a few dimensions of activations reveal anomalous peaks. These anomalous activations are usually related to an expert's specific abilities. Traditional importance metrics, such as the $\mathcal{L}_2$ norm and mean values, only capture global average properties, resulting in an inaccurate characterization of an expert's specialization. To address this challenge, we introduced the Anomalous Contribution Integration (ACI), which can comprehensively evaluate the importance of each collaborative unit from two perspectives: inner energy and downstream influence. Based on the $H$ weighting by $g$, we utilize mean magnitude, magnitude variance, and peak magnitude to form a comprehensive score via a weighted sum. This score is then multiplied by the squared $\mathcal{L}_2$-norm of the corresponding column in the $\mathbf{W}_{down}$ to evaluate the dimension's impact on the output collectively. Furthermore, as for downstream influence, we consider that an anomalous activation must have an impact on both the current layer and the next layer to ensure effective information delivery. Therefore, we approximate this by calculating a weighted alignment score between $\mathbf{W}_{down}$ of the current expert and $\mathbf{W}_{up}$ and $\mathbf{W}_{gate}$ in the next layer. The entire ACI is weighted by both inner energy and downstream influence, creating a robust metric for evaluating the importance of each dimension. Algorithm 1 outlines the concrete pseudocode.

---

**Algorithm 1** ACI: importance scoring and global grouping

---

**Require:** For each expert $e$ in $E$: activations $H^{(e)} \in \mathbb{R}^{N \times D}$, routing weights $r^{(e)} \in \mathbb{R}^N$, down weights $W_{down}^{(e)} \in \mathbb{R}^{H \times D}$; hyperparams $\gamma, w_{mean}, w_{var}, w_{peak}$; layer $l$
1:  $\mathcal{S}_{global} = \emptyset$
2:  **for** each expert $e$ in layer $l$ **do**
3:      $W_{act} \leftarrow H^{(e)} \cdot r^{(e)}$
4:      $S_{hyb} \leftarrow w_{mean}\text{mean}(|W_{act}|^2) + w_{var}\text{var}(|W_{act}|^2) + w_{peak} \max |W_{act}|$
5:      $E_{proj} \leftarrow \|\text{columns of } \mathbf{W}_{down}^{(e)}\|_2^2$
6:      $I_{inner} \leftarrow S_{hyb} \odot E_{proj}$
7:      $V_{out} \leftarrow (\mathbf{W}_{down}^{(e)})^\top; \quad V_{in} \leftarrow \text{Concat}[(\mathbf{W}_{gate}^{(E^{l+1})} + \mathbf{W}_{up}^{(E^{l+1})})/2]$
8:      $A \leftarrow |V_{out}@V_{in}^\top|$
9:      $I_{downstream} \leftarrow A@\|\text{rows of } V_{in}\|_2$
10:     $I \leftarrow I_{inner} + \gamma \cdot \text{norm}(I_{downstream})$
11:     Append scores from $e$ to $\mathcal{S}_{global}$
12: **end for**
13: **return** $\mathcal{S}_{global}$

---

## 3.4 LOW-RANK AND SPARSE APPROXIMATION

Based on the ACI score we calculate, we can globally rank all the dimensions and partition them into two groups: a high-importance group, corresponding to the sparse component, and the remaining dimensions, which are defined as the low-rank component. Each element is processed with a tailored compression strategy to preserve the crucial expertise.

**Sparse Component Preservation.** A few collaborative units identified as high-importance are considered to store professional knowledge, which is crucial for the function of experts. Therefore, to prevent the loss of information, we regard the corresponding rows in $\mathbf{W}_{up}$ and $\mathbf{W}_{gate}$ and columns in $\mathbf{W}_{down}$ as the sparse components, which are preserved in their original form.

**Low-rank Component Approximation.** As shown in Figure 1, after removing the sparse components and applying activation-aware SVD to the expert weights, we can observe that the energy of the matrices is concentrated in a few of the largest singular values. This provides strong evidence to perform a low-rank approximation for the remaining dimensions, rather than simply pruning. To enhance the effectiveness of compression, we adapt an activation-aware SVD method, which is proposed by Wang et al. (2025b).

First, we perform the eigenvalue decomposition on the Gram matrix from the activations $X$ to extract the primary energy of the input features: $E, V = \text{EVD}(X^\top X)$. Using the eigenvectors and eigenvalues, we project the original weights $\mathbf{W}_{lr}$ into activation space and perform SVD on it:

$$U_w S_w V_w^\top = \text{SVD}(\mathbf{W}_{lr} V E^{\frac{1}{2}})$$

Subsequently, after retaining the top $k$ singular values, we obtain the low-rank factors $U_k$, $S_k$, and $V_k$. Then, we reverse the initial transformation, projecting the weight back from the activation space:

$$\mathbf{W}_{lr\_svd} = U_k S_k V_k^\top E^{-\frac{1}{2}} V^\top$$

The final low-rank factors are represented as: $\mathbf{W}_{com} = U_k S_k$, $\mathbf{W}_{rec} = V_k^\top E^{-\frac{1}{2}} V^\top$. Moreover, drawing inspiration from $D^2$-MoE (Gu et al., 2025a), we introduced incremental learning for $\mathbf{W}_{down}$ and obtained a base matrix. In contrast, we adapt ridge regression with a regularization term to incorporate general knowledge and compensate for truncation errors, instead of relying on Fisher information. The objective function for the ridge regression is as follows:

$$\mathcal{J}(\mathbf{B}) = \|Y - H_c \mathbf{B}^\top\|_F^2 + \lambda \|\mathbf{B}\|_F^2$$

where $Y$ is the original output of the expert and $H_c$ is the intermediate output of compressed $\mathbf{W}_{gate}$ and $\mathbf{W}_{up}$. Here, $\|\cdot\|_F$ denotes the Frobenius norm, and $\lambda$ is the regularization coefficient obtained via grid search.

## 3.5 Mutual Information-Guided Layer Compression

Deep neural networks usually exhibit discrepancies in information redundancy across different layers. Shallow layers typically focus on extracting versatile local and low-level features with low redundancy. In contrast, deeper layers primarily produce high-level and abstract features that usually present higher redundancy. It makes them more suitable for compression. Several studies have shown that using a uniform compression ratio across all layers often results in performance degradation (Zhong et al., 2025a; Ding et al., 2025), which motivates the development of a layer-wise allocation method. To accurately evaluate layer-wise redundancy, we propose a method based on mutual information (MI) estimation. In our opinion, if the feature representation of a layer can be inferred from its adjacent layers, its unique contribution is limited, indicating information redundancy. Building upon this, we employ Mutual Information Neural Estimation (MINE) (Belghazi et al., 2018) to estimate mutual information via activation features, thereby capturing complex dependencies between different layers. The specific procedure is as follows: Firstly, we randomly sample some unlabeled texts and feed them into a pre-trained MoE LLMs to acquire the hidden states $Y$ of each layer. We then process these features using masked pooling to obtain the feature encoding $z_i^l$, which represents the $i$-th input sample at layer $l$. Next, by constructing joint samples $(z_k^i, z_k^j)$ and marginal samples $(z_k^i, z_m^j)$, we train an MINE $T(\cdot, \cdot; \theta)$ to approximate the Donsker-Varadhan lower bound for each pair of adjacent layers $(l, l+1)$, thereby estimating the MI between them. The MINE is optimized via the following loss function:

$$\mathcal{L}(\theta) = -\left(\mathbb{E}_{P(Z^l, Z^{l+1})}[T(z^l, z^{l+1}; \theta)] - \log(\mathbb{E}_{P(Z^l)P(Z^{l+1})}[e^{T(z^l, z^{l+1}; \theta)}])\right)$$

where $\mathbb{E}_{P(Z^l, Z^{l+1})}$ and $\mathbb{E}_{P(Z^l)P(Z^{l+1})}$ denote the expectations under the joint and marginal distribution of the layer activations, respectively. Furthermore, we define a redundancy score vector $\mathbf{R} = [R_1, R_2, \ldots, R_n]$ and calculate the score of each layer by average the mutual information with its neighbors. Finally, we formulate a constrained optimization problem to obtain the save ratio $\mathbf{s}_l$ of each layer, aiming to minimize an objective function that balances fidelity to the redundancy scores with inter-layer smoothness. The objective function is defined as follows:

$$L(\mathbf{s}) = L_{\text{fidelity}}(\mathbf{s}) + \lambda_{\text{smooth}} \cdot L_{\text{smooth}}(\mathbf{s}) + \lambda_{\text{reg}} \cdot L_{\text{reg}}(\mathbf{s})$$

where $\lambda_{\text{smooth}}$ and $\lambda_{\text{reg}}$ are hyperparameters which control the smoothness and regularization penalties, respectively. Due to the constraints on the global average sparsity ratio and per-layer bounds, we adapt Quadratic Programming to solve this objective function.

## 4 EXPERIMENTS

In this section, we evaluate our proposed RS-MoE across multiple tasks and compare it with many state-of-the-art MoE compression methods. Additionally, we also conduct ablation studies to analyze the contribution of each component.

| Ratio | Method | Wiki. | PTB | C4 | ARC-e | HellaS. | Math. | Openb. | PIQA | WinoG. | Avg. |
|-------|--------|-------|-----|-----|-------|---------|-------|--------|------|--------|------|
| | | | | | **Deepseek-MoE-16B-base** | | | | | | |
| 0% | Original | 6.51 | 9.74 | 10.20 | 0.77 | 0.58 | 0.32 | 0.33 | 0.79 | 0.72 | 0.59 |
| 20% | NAEE | 7.58 | 13.73 | 14.01 | 0.71 | 0.55 | 0.29 | 0.32 | 0.77 | 0.67 | 0.55 |
| | $D^2$-MoE | 7.02 | 11.56 | 12.62 | 0.74 | 0.54 | 0.31 | 0.30 | 0.75 | 0.69 | 0.56 |
| | RS-MoE | **6.74** | **10.42** | **11.28** | **0.76** | **0.56** | **0.32** | **0.33** | **0.77** | **0.71** | **0.58** |
| 40% | NAEE | 8.57 | 14.41 | 18.12 | 0.67 | 0.41 | 0.26 | 0.23 | 0.70 | 0.67 | 0.49 |
| | $D^2$-MoE | 8.30 | 14.58 | 17.64 | **0.69** | 0.45 | 0.27 | 0.26 | 0.72 | 0.65 | 0.51 |
| | RS-MoE | **8.15** | **13.26** | **14.93** | 0.67 | **0.48** | **0.28** | **0.28** | **0.73** | **0.68** | **0.52** |
| 60% | NAEE | 19.08 | 35.92 | 38.11 | 0.49 | 0.33 | 0.23 | 0.18 | 0.61 | 0.57 | 0.40 |
| | $D^2$-MoE | 12.25 | 27.79 | 30.76 | 0.54 | 0.34 | 0.24 | 0.20 | 0.63 | 0.60 | 0.43 |
| | RS-MoE | **9.95** | **18.29** | **22.52** | **0.59** | **0.40** | **0.26** | **0.26** | **0.68** | **0.65** | **0.47** |
| | | | | | **Mixtral-8×7B** | | | | | | |
| 0% | Original | 3.98 | 14.56 | 7.14 | 0.84 | 0.65 | 0.41 | 0.36 | 0.82 | 0.76 | 0.64 |
| 20% | NAEE | 4.72 | 16.84 | 9.11 | 0.77 | 0.60 | **0.40** | 0.32 | 0.78 | 0.72 | 0.60 |
| | $D^2$-MoE | **4.67** | 16.52 | 8.96 | 0.80 | 0.61 | 0.39 | 0.32 | **0.81** | 0.75 | 0.61 |
| | RS-MoE | 4.70 | **16.49** | **8.52** | **0.81** | **0.62** | 0.39 | **0.33** | 0.80 | **0.75** | **0.62** |
| 40% | NAEE | 6.51 | 21.83 | 13.97 | 0.63 | 0.48 | **0.35** | 0.25 | 0.72 | 0.64 | 0.51 |
| | $D^2$-MoE | 5.97 | 21.66 | **11.87** | 0.78 | 0.54 | 0.33 | 0.29 | 0.77 | **0.71** | 0.57 |
| | RS-MoE | **5.83** | **18.23** | 12.54 | **0.78** | **0.56** | 0.33 | **0.30** | 0.78 | 0.70 | **0.58** |
| 60% | NAEE | 10.84 | 35.23 | 24.17 | 0.51 | 0.38 | 0.27 | 0.19 | 0.62 | 0.58 | 0.43 |
| | $D^2$-MoE | 7.83 | 26.73 | 15.85 | 0.68 | 0.50 | 0.29 | **0.27** | 0.71 | **0.69** | 0.52 |
| | RS-MoE | **7.74** | **23.43** | **15.36** | **0.71** | **0.51** | **0.31** | 0.26 | **0.71** | 0.67 | **0.53** |
| | | | | | **Qwen3-30B-A3B** | | | | | | |
| 0% | Original | 8.65 | 13.41 | 13.17 | 0.78 | 0.69 | 0.58 | 0.42 | 0.79 | 0.70 | 0.66 |
| 20% | NAEE | 8.95 | 14.18 | 13.77 | 0.76 | 0.68 | 0.51 | 0.42 | 0.78 | **0.69** | 0.64 |
| | $D^2$-MoE | 9.12 | 17.64 | 18.28 | 0.73 | 0.64 | 0.49 | 0.41 | 0.76 | 0.66 | 0.62 |
| | RS-MoE | **8.87** | **13.93** | **13.36** | **0.77** | **0.68** | **0.53** | **0.42** | **0.79** | 0.67 | **0.64** |
| 40% | NAEE | 10.07 | 15.28 | **14.93** | 0.70 | 0.63 | 0.44 | **0.40** | 0.75 | 0.65 | 0.60 |
| | $D^2$-MoE | 14.47 | 26.58 | 21.72 | 0.67 | 0.59 | 0.40 | 0.37 | 0.72 | 0.62 | 0.56 |
| | RS-MoE | **9.48** | **15.10** | 15.05 | **0.71** | **0.65** | **0.44** | 0.39 | **0.77** | **0.66** | **0.60** |
| 60% | NAEE | 13.76 | **19.22** | **20.01** | **0.65** | 0.58 | 0.35 | 0.34 | 0.70 | 0.60 | 0.54 |
| | $D^2$-MoE | 21.76 | 38.84 | 36.55 | 0.60 | 0.52 | 0.33 | 0.29 | 0.65 | 0.58 | 0.50 |
| | RS-MoE | **13.56** | 20.17 | 20.12 | 0.63 | **0.60** | **0.39** | **0.34** | **0.71** | **0.61** | **0.55** |

Table 1: Performance comparison of RS-MoE on three mainstream MoE models, with the original model included as a baseline. The best results are marked in bold.

### 4.1 GENERAL SETUP

**Models and Datasets.** To assess the effectiveness of our RS-MoE, we conduct comprehensive experiments on three open-source MoE LLMs: DeepSeekMoE-16B-Base (Dai et al., 2024), Qwen3-30B-A3B (Yang et al., 2025), and Mixtral-8×7B (Jiang et al., 2024a). Regarding datasets, we evaluated our method for two types of tasks: (1) language modeling tasks, including Wikitext2 (Merity et al., 2017), PTB (Marcus et al., 1993), and C4 (Raffel et al., 2020), which are evaluated by perplexity. (2) downstream tasks, including ARC-easy (Clark et al., 2018), HellaSwag (Zellers et al., 2019), MathQA (Amini et al., 2019), OpenbookQA (Mihaylov et al., 2018), PIQA (Bisk et al., 2020), and WinoGrande (Sakaguchi et al., 2020), which are evaluated by accuracy.

**Baseline.** We conducted comparative experiments with three other state-of-the-art methods for MoE compression, including NAEE (Lu et al., 2024), MoE-I$^2$ (Yang et al., 2024) and $D^2$-MoE (Gu et al., 2025a).

**Implementation details.** For all experiments, we randomly sampled 128 samples from the Wikitext2 datasets, which are truncated to a sequence length of 2048 tokens. All experiments were performed on NVIDIA A800 GPUs. Further details can be found in Appendix A.2.

## 4.2 MAIN RESULTS

As shown in Table 1, we conducted a comprehensive comparison of our RS-MoE against three state-of-the-art methods under different sparsity ratios. Experimental results demonstrate that RS-MoE achieves outstanding performance across different baselines, tasks, and sparsity ratios. In particular, under a 20% sparsity ratio, RS-MoE achieves a perplexity (PPL) of 9.48 in language modeling tasks and a downstream task accuracy of 58% with Deepseek-MoE-16B-base, surpassing other methods. Notably, as the sparsity ratio increases, the performance of our method becomes more remarkable than that of other methods. For instance, when the sparsity ratio increases from 20% to 60%, the performance degradation of RS-MoE increases from 2% to 20%, whereas that of $D^2$-MoE escalates from 3.5% to 27%. This strong performance extends to larger models. For the Qwen3-30B-A3B at 20% sparsity, RS-MoE attains a PPL of 8.87, nearly matching the original model's 8.65, while maintaining a competitive accuracy of 64%.

## 4.3 ABLATION STUDY

**Collaborative Decomposition.** To further validate the effectiveness of our collaborative decomposition, we calculate the ACI of $\mathbf{W}_{gate}$, $\mathbf{W}_{up}$, and $\mathbf{W}_{down}$ independently. As shown in Table 2, under different compression ratios, the perplexity and precision of our collaborative decomposition always perform better than compressing each matrix independently. The result demonstrates that our framework effectively leverages the correlations among expert matrices, thereby reducing the parameters while minimizing the loss of local information.

| Ratio | Method | Wiki. | PTB | C4 | ARC-e | HellaS. | Math. | Openb. | PIQA | WinoG. | Avg. |
|-------|--------|-------|-----|-----|-------|---------|-------|--------|------|--------|------|
| 0% | Original | 6.51 | 9.74 | 10.20 | 0.77 | 0.58 | 0.32 | 0.33 | 0.79 | 0.72 | 0.59 |
| 20% | Independence | 7.17 | 11.13 | 12.03 | 0.74 | 0.54 | 0.31 | 0.33 | 0.77 | 0.70 | 0.57 |
| | Collaboration | **6.74** | **10.42** | **11.28** | **0.76** | **0.56** | **0.32** | **0.33** | **0.77** | **0.71** | **0.58** |
| 40% | Independence | 8.38 | 13.70 | 15.42 | 0.66 | 0.47 | 0.26 | 0.27 | 0.70 | 0.67 | 0.51 |
| | Collaboration | **8.15** | **13.26** | **14.93** | **0.67** | **0.48** | **0.28** | **0.28** | **0.73** | **0.68** | **0.52** |
| 60% | Independence | 10.97 | 19.36 | 23.91 | 0.58 | 0.39 | 0.24 | 0.21 | 0.66 | 0.64 | 0.45 |
| | Collaboration | **9.95** | **18.29** | **22.52** | **0.59** | **0.40** | **0.26** | **0.26** | **0.68** | **0.65** | **0.47** |

Table 2: Performance comparison between collaborative decomposition and independent decomposition based on DeepSeekMoE-16B-Base, with the original model included as a baseline. The best results are marked in bold.

**Layerwise Sparsity Allocation.** We investigate the effects of layerwise sparsity on model performance. Specifically, we adjust the parameters of different layers to maintain a fixed sparsity ratio. Table 3 reveals the result with a different proportion. It can be observed that our method, which assigns lower sparsity to lower layers and higher sparsity to higher layers, outperforms both uniform allocation and the inverse strategy (i.e., higher sparsity for lower layers, lower sparsity for higher layers).

**Effectiveness of Sparse and Low-Rank Components.** Table 4 compares RS-MoE with structured pruning, standard SVD, and activation-aware SVD. To ensure a fair comparison, a consistent workflow was applied to all methods, resulting in a 60% compression ratio on the Deepseek-MoE-16B-base. The experimental results reveal that RS-MoE consistently outperforms structured pruning strategies, reducing the PPL by approximately 10%. Additionally, the activation-aware SVD outperforms the Standard SVD because it contains feature information.

Table 3: Results of different sparsity allocation.

| Strategy | Wikitext-2 | PTB | C4 | Average |
|---|---|---|---|---|
| Uniform | 8.12 | 13.59 | 15.20 | 12.30 |
| Reverse | **8.10** | 13.94 | 15.46 | 12.50 |
| RS-MoE | 8.15 | **13.26** | **14.93** | **12.11** |

Table 4: Comparison of compression strategies

| Methods | Wikitext-2 | PTB | C4 | Average |
|---|---|---|---|---|
| Original | 6.51 | 9.74 | 10.20 | 8.82 |
| Pruning | 10.27 | 19.71 | 24.59 | 18.19 |
| Standard SVD | 10.23 | 19.37 | 24.10 | 17.90 |
| RS-MoE | **9.95** | **18.29** | **22.52** | **16.92** |

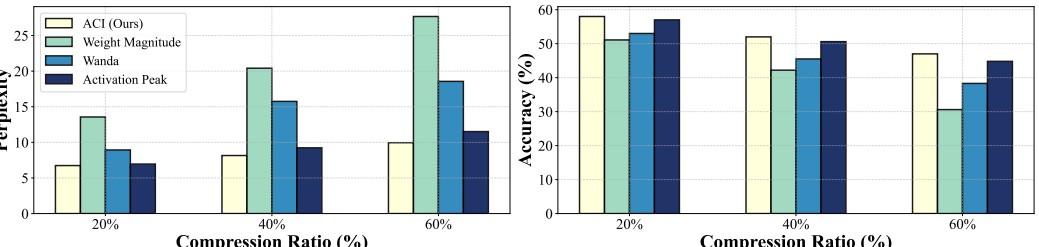

Figure 3: Comparison results of Deepseek-MoE-16B-base.

**Comparison of Grouping Metrics.** We show the impact of different grouping metrics on the LLM's perplexity in Figure 3. The evaluation was conducted across various sparsity ratios, comparing four metrics: ACI, weight magnitude, activation magnitude (as used in Wanda (Sun et al., 2024)), and activation peak. It can be concluded that ACI can effectively identify the critical parts of the weights, resulting in a decrease in compression error. For instance, the PPL of ACI is about 17 points lower than that of the common weight magnitude method. Compared to the simple activation peak, it remains approximately 20% lower. Otherwise, as the sparsity increases, the benefit of ACI is particularly pronounced.

**Robustness to Calibration Samples.** We attempt a different number of calibration samples, ranging from 8 to 256. As revealed in Figure 4, compared with the $D^2$-MoE, RS-MoE is more robust when only a few calibration samples are provided.

**Base Weight Construction.** In this experiment, we validate the effectiveness of the proposed method for constructing the base matrix, which is based on ridge regression. Table 5 compared our method with the following merging approaches: Fisher merging (Matena & Raffel, 2022), frequency merging, mean value merging, TIES (Yadav et al., 2023) and PCB (Du et al., 2024). Obviously, although both Fisher merging (PPL 18.31) and frequency merging (PPL 23.03) achieve high performance, the ridge regression approach more effectively compensates for the error between the actual and compressed outputs, achieving superior performance (PPL 16.92).

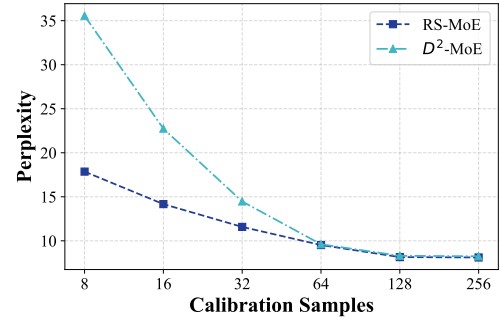

Figure 4: Impact of calibration samples.

Table 5: Results of different Base Weights.

| Methods | WikiText-2 | PTB | C4 | Average |
|---|---|---|---|---|
| Mean | 13.74 | 30.18 | 35.83 | 26.58 |
| Frequency | 12.83 | 26.54 | 29.72 | 23.03 |
| PCB | 17.85 | 39.56 | 46.94 | 34.78 |
| TIES | 23.38 | 51.64 | 71.27 | 48.76 |
| Fisher | 10.34 | 19.75 | 24.84 | 18.31 |
| **Ours** | **9.95** | **18.29** | **22.52** | **16.92** |

Table 6: Efficiency analysis of RS-MoE

| Ratio | Cost of Time (ms) | | |
|---|---|---|---|
| | Deepseek | Mixtral | Qwen |
| 0% | 2.11 | 43.99 | 1.22 |
| 20% | 1.99 (1.06×) | 33.19 (1.33×) | 1.24 (0.98×) |
| 40% | 1.43 (1.48×) | 25.96 (1.69×) | 1.07 (1.14×) |
| 60% | 1.14 (1.85×) | 17.67 (2.49×) | 0.65 (1.88×) |

## 4.4 Efficiency Analysis

Each expert network consists of three components: $\mathbf{W}_{gate}$, $\mathbf{W}_{up}$ and $\mathbf{W}_{down}$. We selected a sequence length of 2048 to measure the latency of matrix multiplication for three models at various compression rates, both before and after decomposition. Table 6 presents the average time consumption in milliseconds and the corresponding speedup ratios after 500 iterations. For Mixtral-8x7B, RS-MoE accelerates the matrix multiplication speed by $1.33\times$ at a 20% compression ratio. This speedup further increases to $2.49\times$ at a 60% compression ratio. The result demonstrates that RS-MoE effectively accelerates computation by replacing weight matrices with smaller ones and leveraging existing hardware capabilities. More details can be found in Appendix A.5.

## 5 Conclusion

In this paper, we introduce RS-MoE, a novel compression framework tailored for MoE LLMs, specifically designed to mitigate the substantial storage and memory challenges inherent to these models. Our approach is built upon the key observation that an expert's weights can be collaboratively decomposed into two components: a sparse component capturing critical, specialized knowledge, and a low-rank component representing more general features. By leveraging the sparse structure within intermediate activation peaks, our method collaboratively decomposes the expert weights into these sparse and low-rank components, thus maintaining the integrity and specialized functionality of each expert. Our framework systematically integrates several techniques to achieve efficient and performance-preserving compression. These include a comprehensive importance score (ACI) based on activation peaks to guide the decomposition, a mutual information-based strategy for layer-wise sparsity allocation, and activation-aware SVD combined with ridge regression to minimize reconstruction errors. Extensive experiments on models such as Deepseek-MoE-16B-base, Mixtral-8x7B, and Qwen3-30B-A3B demonstrate that RS-MoE consistently outperforms state-of-the-art methods across various downstream tasks, especially at high compression ratios.

## Ethics Statement

This research strictly adheres to the ICLR Code of Ethics. The research process involved no human or animal experiments, and no personally identifiable information was used. All datasets were handled in compliance with their terms of use and privacy policies. We are committed to mitigating bias and discrimination in our methodology and ensuring the transparency and integrity of our work.

## Reproducibility Statement

To ensure our results are fully reproducible, we have included our code in the supplementary materials. This paper provides a detailed description of the experimental setup, covering model configurations, training procedures, and the hardware environment. To facilitate replication, a comprehensive explanation of our core contribution is also included. Our evaluation process relies on public datasets, such as Wikitext2, to ensure consistent benchmarking.

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

## A APPENDIX

### A.1 STATEMENT ON LLM USAGE

In accordance with the ICLR 2026 policies concerning the utilization of LLMs, it is hereby disclosed that the exclusive function of LLMs in this work was to provide writing assistance in the preparation of this manuscript. Specifically, we employed Gemini exclusively for language polishing, including improving grammatical accuracy and enhancing sentence clarity and readability.

It is emphasized that all research ideas, methodologies, experimental designs, and scientific contributions presented in this paper are original work by the authors. The experimental results, data analysis, and conclusions were produced entirely by the authors without any assistance from an LLM. The utilization of Gemini was strictly constrained to enhancing the linguistic exposition of our research findings, without impacting or contributing to the technical content or scientific merit of this work.

The authors accept full responsibility for all content presented in this submission, including the accuracy of all claims, the validity of experimental results, and the appropriateness of conclusions drawn.

## A.2    IMPLEMENTATION DETAIL

In this section, we provide the detailed implementation of our RS-MoE framework to ensure the reproducibility of our experiments.

All experiments were conducted on NVIDIA A800 GPUs using core libraries such as PyTorch, Transformers, and Datasets. We utilized the torch.bfloat16 data type for all model weights and computations to strike a balance between precision and efficiency.

**Calibration and Feature Collection.** For all models, we performed calibration using 128 samples randomly selected from the Wikitext2 training dataset, with the random seed set to 42 for consistency. Each sample was truncated to a sequence length of 2048 tokens.

**Anomalous Contribution Integration (ACI).** The ACI score, which is central to our method, is calculated with specific hyperparameters to identify critical collaborative units robustly. The score is a composite of inner energy and downstream influence. The inner energy component is a weighted sum of normalized mean energy ($w_{mean} = 0.4$), variance of energy ($w_{var} = 0.05$), and peak activation magnitude ($w_{peak} = 0.8$). The downstream influence, which measures the alignment with subsequent layers, is incorporated with a weighting factor of $\gamma = 0.05$. These parameters were determined through empirical validation to distinguish specialized knowledge from general features effectively.

**Low-Rank and Sparse Approximation.** Our collaborative decomposition strategy is guided by the ACI scores and a layer-wise sparsity ratio derived from Mutual Information Neural Estimation (MINE). High-importance units are preserved in their original form. Medium-importance units undergo activation-aware SVD, where the rank is dynamically determined based on the allocated parameter budget for that expert group, aiming to retain essential information while maximizing compression. Low-importance units are structurally pruned by setting their corresponding weights to zero. For the $\mathbf{W}_{down}$, we employ ridge regression to learn a shared base weight that compensates for global reconstruction error, with a regularization parameter of $\lambda = 1e\text{-}3$. The final compressed model is instantiated by replacing the original MoE layers with a highly optimized custom module that efficiently reconstructs expert outputs from the preserved sparse components and low-rank factors during inference.

## A.3    LAYER-WISE PARAMETER BUDGET

To achieve efficient compression, we leverage MINE to evaluate the redundancy of each MoE layer and dynamically allocate parameter budgets accordingly. In principle, layers with higher mutual information are considered more redundant and are thus assigned a smaller parameter budget. We compute the final budget allocation using a Quadratic Programming (QP) solver with a smoothness constraint. Figure 5 below illustrates the parameter budget allocated by our method to each MoE layer across three overall compression ratios (20%, 40%, and 60%). It clearly shows that the parameter budget allocation exhibits a complex, fluctuating pattern, rather than a simple monotonic decrease with layer depth. For instance, there are noticeable budget drops around layers 5 and 20, and a significant peak around layer 9. Importantly, this allocation pattern remains highly consistent across the different overall sparsity ratios, demonstrating that our method can stably identify the relative importance of different layers within the model. Meanwhile, as the overall sparsity ratio increases (from 0.2 to 0.6), the parameter budget for all layers is reduced proportionally.

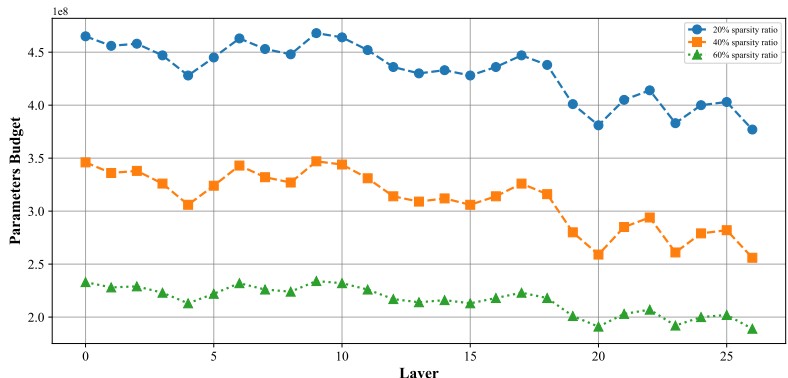

Figure 5: Parameter budget of each layer for Deepseek-MoE-16B-base.

## A.4 ANALYSIS OF SPARSITY AND LOW-RANK PROPERTIES IN EXPERTS FROM DIFFERENT LAYERS

In Figure 1, we motivated our RS-MoE method by illustrating the sparsity of activation and the low-rank weight structure of a representative expert (Expert 35 in Layer 27). To demonstrate that these properties are not isolated cases but are intrinsic to the model architecture, we provide a comprehensive statistical analysis of *all* experts in Layers 1, 9, 18, and 27 in this section.

**Data Preprocessing for Visualization.** We observed significantly high activation magnitudes in specific experts: Experts 14 and 43 in Layer 1, and Experts 54 and 62 in Layer 27. To prevent these extreme outliers from skewing the vertical scale and obscuring the distribution details of other experts, we clipped the top 2% of the activation values for these specific experts in the visualization.

**Distribution Analysis.** As evidenced by the consistent patterns across the four analysed layers, the majority of experts exhibit a significant concentration of energy within the top singular values (indicated by the rapid transition from dark to light colours in the heatmaps). Meanwhile, their activation statistics exhibit a highly skewed distribution: a small subset of neurons receives strong activation, whereas the vast majority retain negligible magnitude. This universality strongly supports the robustness of the activation sparsity and low-rank assumption underlying our proposed method.

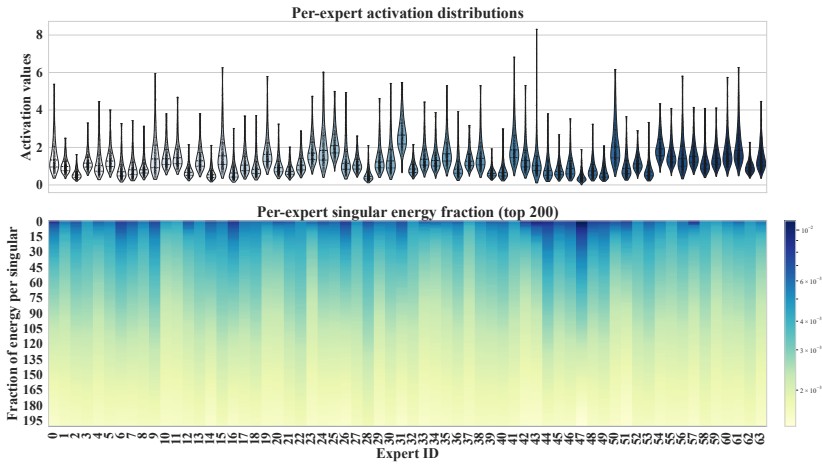

Figure 6: Activation and singular value distribution for all experts in Layer 1.

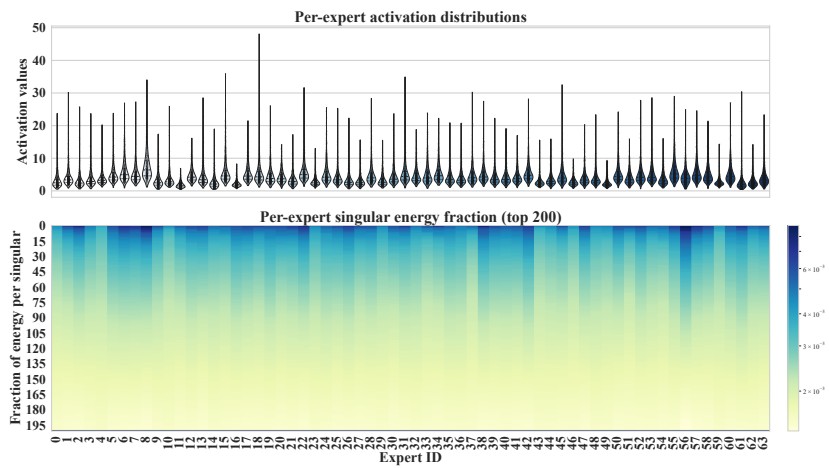

Figure 7: Activation and singular value distribution for all experts in Layer 9.

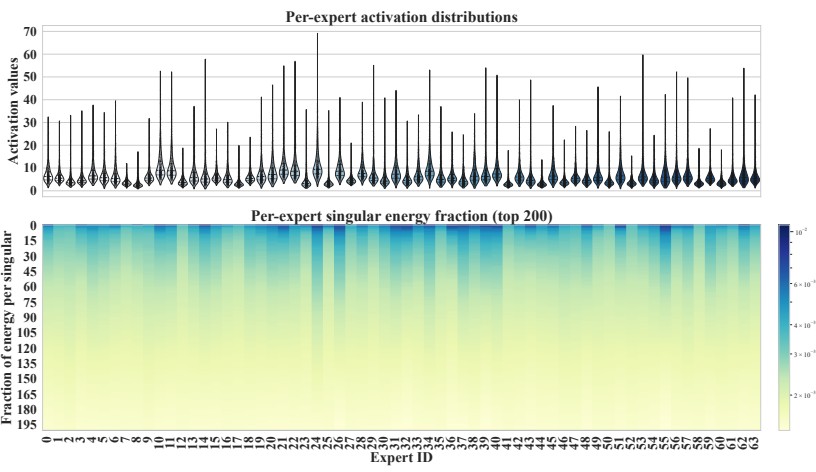

Figure 8: Activation and singular value distribution for all experts in Layer 18.

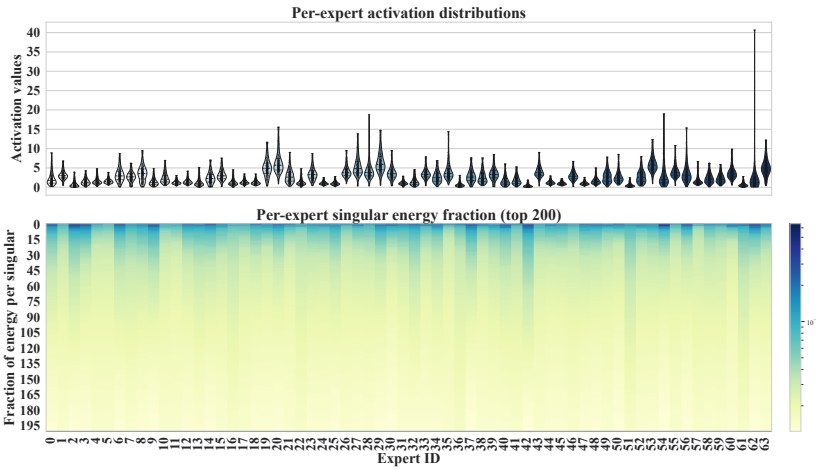

Figure 9: Activation and singular value distribution for all experts in Layer 27.

## A.5 COMPUTATIONAL COST DISCUSSION OF RS-MoE

In this section, we provide more details about offline compression cost and online inference efficiency. The RS-MoE process consists of three stages. First, MINE estimates mutual information and allocates layerwise compression ratios, a step that takes approximately 2 minutes to set the compression ratio and save a cache file when running on DeepSeekMoE-16B. Next, we calculate ACI based on activations to provide evidence for decomposition; this requires about 6 minutes. Finally, we slice and perform SVD on the expert matrix using ACI, with both operations together taking 24 minutes. These steps collectively describe the computational cost and efficiency of our workflow. Computational cost of other models is shown in Table 7.

Table 7: **Offline Compression Cost.** Compressing time and memory used of different models

| Stage | Metric | DeepSeekMoE-16B | Mixtral-8x7B | Qwen3-30B-A3B |
|---|---|---|---|---|
| **MINE** | Time Cost | 2 mins | 5 mins | 4 mins |
| | Peak VRAM | 35.72 GB | 122.47 GB | 72.02 GB |
| **ACI** | Time Cost | 6 mins | 9 mins | 8 mins |
| | Peak VRAM | 34.86 GB | 121.56 GB | 71.37 GB |
| **Slice & SVD** | Time Cost | 24 mins | 61 mins | 43 mins |
| | Peak VRAM | 42.82 GB | 125.67 GB | 71.58 GB |

Furthermore, we chose a sequence length of 2048 to evaluate the online inference efficiency of our method. Specifically, we conducted 500 iterations on NVIDIA A800 GPUs using float32 precision to measure the average matrix multiplication runtime for $\mathbf{W}_{gate}$, $\mathbf{W}_{up}$, and $\mathbf{W}_{down}$ across various models and compression ratios. The results are presented in Table 8.

Table 8: **Online Inference Efficiency.** Matrix multiplication runtime of different components.

| Model | Ratio | Operations(ms) | | | Total (speedup) |
|---|---|---|---|---|---|
| | | Gate | Up | Down | |
| **DeepSeekMoE-16B** | 0% | 0.69 | 0.69 | 0.73 | 2.11 |
| | 20% | 0.65 | 0.65 | 0.69 | 1.99 (1.06×) |
| | 40% | 0.47 | 0.47 | 0.49 | 1.43 (1.48×) |
| | 60% | 0.36 | 0.37 | 0.41 | 1.14 (1.85×) |
| **Mixtral-8x7B** | 0% | 13.35 | 14.09 | 16.55 | 43.99 |
| | 20% | 10.96 | 10.88 | 11.35 | 33.19 (1.33×) |
| | 40% | 8.09 | 8.12 | 9.75 | 25.96 (1.69×) |
| | 60% | 5.65 | 5.81 | 6.21 | 17.67 (2.49×) |
| **Qwen3-30B-A3B** | 0% | 0.39 | 0.40 | 0.43 | 1.22 |
| | 20% | 0.40 | 0.40 | 0.44 | 1.24 (0.98×) |
| | 40% | 0.34 | 0.36 | 0.37 | 1.07 (1.14×) |
| | 60% | 0.21 | 0.21 | 0.23 | 0.65 (1.88×) |

## A.6 HYPERPARAMETER SENSITIVITY ANALYSIS

To validate the robustness of our proposed ACI score, we conducted a sensitivity analysis on its most critical hyperparameter, $w_{peak}$. We evaluated perplexity on the WikiText-2 dataset using the DeepSeekMoE-16B model at 60% sparsity, varying $w_{peak}$ from 0.4 to 1.2. As illustrated in Figure 10, the best performance is achieved around our default setting of $w_{peak} = 0.8$. In addition, within the robustness zone, RS-MoE maintains high performance even as $w_{peak}$ fluctuates between 0.9 and 1.3. It confirms the importance of activation peaks. The slight degradation at lower values ($w_{peak} < 0.5$) further underscores the necessity of prioritizing peak activations in the importance scoring mechanism.

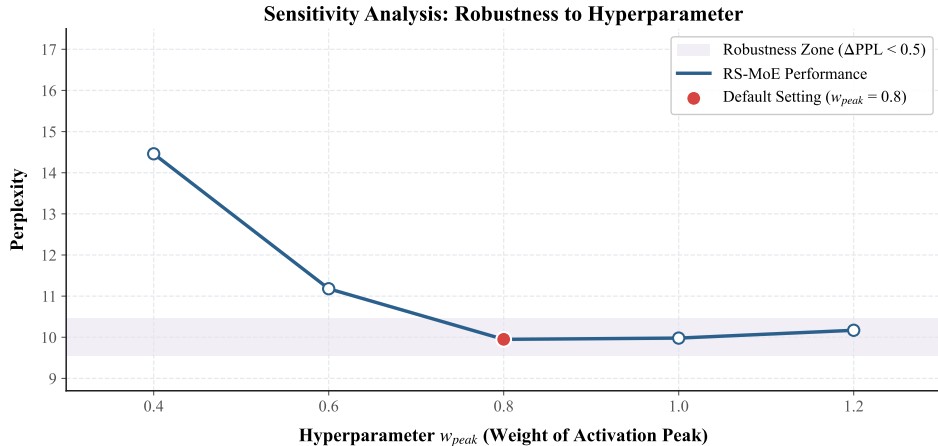

Figure 10: The perplexity on Wikitext-2 using DeepSeekMoE-16B (60% sparsity) as $w_{peak}$ varies. The "Robustness Zone" indicates stable performance.

We further decouple the contribution of inner activation statistics and downstream influence. "Full ACI" represents our proposed method. As revealed in Table 9, the results confirm that incorporating downstream influence ($\gamma$) further improves performance, while activation peaks are the most critical factor.

Table 9: Ablation Study of ACI Components on DeepSeekMoE-16B (60% Sparsity).

| Method Variant | Inner Stats ($w_{peak}$) | Downstream ($\gamma$) | PPL (Wikitext2) |
|---|:---:|:---:|:---:|
| **Full ACI (Ours)** | ✓ | ✓ | **9.95** |
| w/o Downstream | ✓ | ✗ | 10.12 |
| w/o Peak (Mean+Var only) | ✗ | ✓ | 15.76 |
| Weight Magnitude | ✗ | ✗ | 21.35 |

## A.7 EVALUATION ON GENERATIVE SUMMARIZATION TASKS

To further investigate whether RS-MoE preserves the model's ability to generate coherent, accurate long-form text, we experimented on the CNN/DailyMail summarization dataset (Hermann et al., 2015) using DeepSeekMoE-16B. We assessed performance across varying sparsity levels (20%, 40%, and 60%) against the uncompressed original model (0% sparsity). The result is revealed in Table 10. It can be seen that although ROUGE-1, ROUGE-2, and ROUGE-L scores (Lin, 2004) decrease due to compression, they retain a substantial degree of their generative quality.

Table 10: ROUGE scores on CNN/DailyMail for DeepSeekMoE-16B.

| Method | Ratio | CNN/DailyMail (ROUGE) | | |
|---|---|---|---|---|
| | | R-1 | R-2 | R-L |
| Original Model | 0% | 21.81 | 6.88 | 16.00 |
| RS-MoE | 20% | 18.09 | 4.47 | 13.48 |
| RS-MoE | 40% | 17.56 | 4.24 | 13.19 |
| RS-MoE | 60% | 15.76 | 3.98 | 12.75 |

## A.8 ANALYSIS OF EXPERT ROUTING CONSISTENCY

To validate whether RS-MoE might lead to mode collapse or alter the intrinsic routing logic, we conducted both qualitative and quantitative analyses of the expert utilisation distribution on the Wikitext2 dataset.

**Qualitative Visualization.** Figure 11 reveals expert activation frequency across all layers. Comparing the original DeepseekMoE-16B with RS-MoE (60% sparsity), the heatmaps exhibit highly consistent patterns. There is no obvious sign of mode collapse, which would manifest as single-expert dominance.

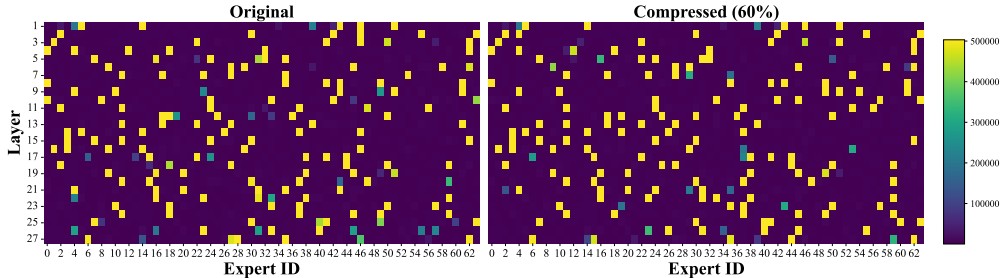

Figure 11: Comparison of expert activation frequency before and after compression.

**Quantitative Metric (Entropy).** We further quantified load balance using expert utilization entropy, calculated as $H = -\sum_{i=1}^{N} p_i \log p_i$, where $N$ is the number of experts and $p_i$ is the utilization frequency of the $i$-th expert. The average entropy of the original model is 2.1415, whereas RS-MoE maintains a comparable value of 2.1273. It confirms that RS-MoE effectively preserves the diversity of expert selection and maintains the router's decision boundaries even at high compression rates.

## A.9 ABLATION OF LOW-RANK RANK SELECTION

To isolate the benefit of our rank selection, we detail the energy-aware allocation. Unlike fixed-rank methods, we distribute the low-rank budget $B_{lr}^{(l)}$ among expert matrices $\mathbf{W}_{gate}, \mathbf{W}_{up}$, and $\mathbf{W}_{down}$ based on spectral complexity.

We first compute the target rank $r_{99}^{(m)}$ required to capture 99% of the activation-weighted spectral energy for each matrix $m$:

$$r_{99}^{(m)} = \min \left\{ k : \frac{\sum_{i=1}^{k} \sigma_i^2}{\sum_j \sigma_j^2} \geq 0.99 \right\}$$

The actual rank $k^{(m)}$ is then allocated proportionally: $k^{(m)} \propto r_{99}^{(m)} \times B_{lr}^{(l)}$. This prioritizes matrices with slower spectral decay. As shown in Table 11, this adaptive strategy outperforms the fixed-rank baseline.

| Rank Allocation Strategy | PPL |
|---|---|
| Standard SVD (Fixed Rank) | 17.03 |
| **RS-MoE (Energy-based Adaptive)** | **16.92** |

Table 11: Ablation study of rank allocation strategies on WikiText-2. Our energy-based adaptive strategy significantly outperforms the fixed-rank baseline.

This strategy ensures matrices with slower spectral decay receive a larger share of the budget. As shown in Table 11, our adaptive approach achieves a PPL of 16.92, outperforming the 17.90 PPL of standard fixed-rank decomposition. This confirms that respecting distinct spectral characteristics is vital for performance.

