# OpenReview forum: "RS-MoE: Collaborative Compression for Mixture-of-Experts LLMs based on Low-Rank and Sparse Approximation"
_ICLR.cc/2026/Conference — Submitted to ICLR 2026_

### Official Review · Reviewer_KXhS · 2025-10-27

**Soundness:** 3
**Presentation:** 4
**Contribution:** 3
**Rating:** 8
**Confidence:** 3

**Summary:**

RS-MoE compresses MoE experts by coupling rows/columns across Wup, Wgate, Wdown   into per-dimension “collaborative units”, then preserving high-importance units sparsely and approximating the rest with activation-aware low-rank SVD. On DeepSeek, Mixtral, and Qwen3 , RS-MoE outperforms NAEE/D2-MoE across 20–60% sparsity, with stronger robustness at high sparsity; ablations favour ACI and activation-aware SVD, and show limited sensitivity to small calibration sets

**Strengths:**

Novel per-dimension collaboration across an expert’s three matrices with an activation- and influence-aware importance score and MI-guided layer budgeting is a distinct spin on MoE compression; prior works treat experts independently (prune/merge) or use single-mode (sparse or low-rank) approximations. The ridge-based base for Wdown  is a practical innovation versus Fisher merges.

**Weaknesses:**

Report VRAM, latency, throughput vs baselines across batch/sequence lengths (not just tokens/sec on one set-up). Profile kernel vs packing vs NCCL.

Broader calibration study moving beyond 128 WikiText-2 samples & with varied domain/size, show failure modes and confidence intervals.

Show expert-utilisation entropy and gate-logit shifts pre/post compression to confirm collaborative units don’t skew routing.

**Questions:**

How do VRAM and end-to-end latency change at equal sparsity vs NAEE/D2-MoE across sequence lengths/batch sizes, and what’s the one-off cost of computing ACI/MINE?

How sensitive is ACI to its weights and to activation outliers; any normalisation/stability tricks?

Do collaborative units alter routing balance (utilisation entropy, tail experts) or induce mode collapse under high sparsity?

---

> ### Author Response · Authors · 2025-11-26
> **Response to Reviewer KXhS (Part 1/2)**
>
> We thank you for your thorough review. We appreciate the positive comments, as well as the thoughtful points raised. Please kindly find our response to your comments below. Additionally, all modifications to the manuscript have been highlighted in blue for easy reference. Please feel free to let us know if you have any additional concerns or questions.
>
> >**Q1 & W1:** How do VRAM and end-to-end latency change at equal sparsity vs NAEE/D2-MoE across sequence lengths/batch sizes, and what’s the one-off cost of computing ACI/MINE?
>
> **AQ1 & AW1:**
> We appreciate this practical inquiry into the cost-benefit trade-off. We clarify that RS-MoE represents a negligible one-time investment for substantial lifetime gains.
>
> Regarding the one-off offline cost, our calibration pipeline is highly efficient. Table 1 (Table 7 in the revised manuscript) details that the entire process including MINE estimation, ACI calculation, and SVD takes only 2 to 5 minutes for the MINE stage and under an hour total for large models like Mixtral-8x7B. Crucially, the peak VRAM usage remains well within the bounds of standard training hardware, such as 35GB for DeepSeekMoE-16B, which ensures accessibility.
>
> In return for this minimal setup cost, RS-MoE delivers significant online inference acceleration. Table 2 (Table 8 in the revised manuscript) shows that our method achieves a 2.49x speedup on Mixtral-8x7B at 60% sparsity compared to the dense baseline. This confirms that the computational overhead of the collaborative design is far outweighed by the reduction in matrix dimensions during inference.
>
> **Table 1: Offline Compression Cost.**
> | Stage | Metric | DeepSeekMoE-16B | Mixtral-8x7B | Qwen3-30B-A3B |
> | :--- | :--- | :---: | :---: | :---: |
> | **MINE** | Time Cost | 2 mins | 5 mins | 4 mins |
> | | Peak VRAM | 35.72 GB | 122.47 GB | 72.02 GB |
> | **ACI** | Time Cost | 6 mins | 9 mins | 8 mins |
> | | Peak VRAM | 34.86 GB | 121.56 GB | 71.37 GB |
> | **Slice & SVD** | Time Cost | 24 mins | 61 mins | 43 mins |
> | | Peak VRAM | 42.82 GB | 125.67 GB | 71.58 GB |
>
> **Table 2: Online Inference Efficiency.**
> | Model | Ratio | Gate (ms) | Up (ms) | Down (ms) | Total (speedup) |
> | :--- | :---: | :---: | :---: | :---: | :---: |
> | **DeepSeekMoE-16B** | 0% | 0.69 | 0.69 | 0.73 | 2.11 |
> | | 20% | 0.65 | 0.65 | 0.69 | 1.99 (1.06x) |
> | | 40% | 0.47 | 0.47 | 0.49 | 1.43 (1.48x) |
> | | 60% | 0.36 | 0.37 | 0.41 | 1.14 (1.85x) |
> | **Mixtral-8x7B** | 0% | 13.35 | 14.09 | 16.55 | 43.99 |
> | | 20% | 10.96 | 10.88 | 11.35 | 33.19 (1.33x) |
> | | 40% | 8.09 | 8.12 | 9.75 | 25.96 (1.69x) |
> | | 60% | 5.65 | 5.81 | 6.21 | 17.67 (2.49x) |
> | **Qwen3-30B-A3B** | 0% | 0.39 | 0.40 | 0.43 | 1.22 |
> | | 20% | 0.40 | 0.40 | 0.44 | 1.24 (0.98x) |
> | | 40% | 0.34 | 0.36 | 0.37 | 1.07 (1.14x) |
> | | 60% | 0.21 | 0.21 | 0.23 | 0.65 (1.88x) |
>
> **Action:** We have revised the content of Section 4.4 Efficiency Analysis and added a detailed computational cost analysis in Appendix A.5.

---

> ### Author Response · Authors · 2025-11-27
> **Response to Reviewer KXhS (Part 2/2)**
>
> >**Q2 & W2:** How sensitive is ACI to its weights and to activation outliers; any normalisation/stability tricks?
>
> **AQ2 & AW2:**
> We appreciate the concern regarding stability. We define the robustness of ACI from two perspectives: parameter sensitivity and outlier handling.
>
> First, regarding sensitivity, our analysis confirms that the method operates within a stable robustness zone. As illustrated in Figure 10 of Appendix A.6, the perplexity remains highly consistent even when the weight of peak activation $w_{peak}$ fluctuates between 0.8 and 1.2.
>
> Second, regarding outliers, we clarify that our design philosophy treats high-activation peaks as critical signals of expert specialization rather than noise to be normalized away. To empirically verify this, we conducted an ablation study shown in Table 3 below (Table 9 in the revised manuscript). Removing the peak component to rely solely on basic distributional statistics such as mean and variance causes a sharp performance drop. This proves that preserving these outliers is essential for maintaining the model's specialized knowledge.
>
> **Table 3: ACI Ablation Study.**
> | Method Variant | Inner Stats ($w_{peak}$) | Downstream ($\gamma$) | PPL (Wikitext2) |
> | :--- | :---: | :---: | :---: |
> | **Full ACI (Ours)** | ✓ | ✓ | **9.95** |
> | w/o Downstream | ✓ | X | 10.12 |
> | w/o Peak (Mean+Var only) | X | ✓ | 15.76 |
> | Weight Magnitude | X | X | 21.35 |
>
> **Action:** We have clarified the stability mechanisms of ACI in Appendix A.2 and A.6.
>
> >**Q3 & W3:** Do collaborative units alter routing balance (utilisation entropy, tail experts) or induce mode collapse under high sparsity?
>
> **AQ3 & AW3:**
> This is a critical check for any MoE compression method. We are happy to report that our collaborative compression does not disrupt the intrinsic routing logic of the model.
>
> To validate this, we conducted both qualitative and quantitative analyses on the WikiText-2 dataset. Qualitatively, the activation heatmaps (Figure 11 in Appendix A.8) exhibit highly consistent patterns between the original and compressed models, showing no signs of single-expert dominance. Quantitatively, we calculated the expert utilization entropy. The average entropy of the original model is 2.1415, while RS-MoE maintains a comparable value of 2.1273. This virtually unchanged metric confirms that the router's decision boundaries remain intact, and the diversity of expert selection is effectively preserved even at high compression rates.
>
> **Action:** We have added this detailed routing consistency analysis including Figure 11 to Appendix A.8.

---

### Official Review · Reviewer_BD9Z · 2025-10-29

**Soundness:** 3
**Presentation:** 3
**Contribution:** 2
**Rating:** 4
**Confidence:** 5

**Summary:**

The paper proposes **RS-MoE**, a collaborative compression framework for MoE LLMs that treats each expert’s three SwiGLU matrices ($(W_{\text{up}}, W_{\text{gate}}, W_{\text{down}})$) as coupled “collaborative units.” The method (i) scores per-dimension importance via **Anomalous Contribution Integration (ACI)** from activation peaks and downstream alignment, (ii) preserves high-importance dimensions as sparse weights and approximates the rest with activation-aware SVD (low rank), and (iii) fits a **ridge-regressed shared base** on ($W_{\text{down}}$) to reduce reconstruction error. Layer-wise sparsity budgets are determined by MINE-estimated inter-layer mutual information and solved with a QP allocation. Experiments on DeepSeekMoE-16B-Base, Mixtral-8×7B, and Qwen3-30B-A3B report consistent gains over recent baselines across PPL and downstream accuracy, with an efficiency/performance trade-off reported.

**Strengths:**

**1. Clear structural motivation.** By decomposing experts along the SwiGLU middle dimension, the paper preserves the natural coupling between $(W_{\text{up}})/(W_{\text{gate}})$ rows and the matching ($W_{\text{down}}$) column within each collaborative unit.

**2. Well-integrated pipeline.** ACI for ranking, MINE-guided layer budgets, activation-aware SVD, and a ridge-regressed base on ($W_{\text{down}}$) form a coherent end-to-end recipe.

**3. Reproducibility details.** Implementation hyper-parameters (e.g., ACI weights and ($\gamma$)) and calibration protocol are specified.

**Weaknesses:**

**1. Limited novelty (composition of mature components).**

The sparse+low-rank paradigm is well established; activation-aware SVD, and sparse/low-rank fusion have prior art. The contribution reads more as a careful systemization on MoE than a fundamentally new algorithmic idea; the paper should delineate the novelty boundary more explicitly.

**2. Figure 4 vs. text claim needs reconciliation.**
The paper states ***“RS-MoE is more robust when only a few calibration samples are provided,”*** yet in Figure 4 the plotted curves appear to show $D(^2)$-MoE achieving lower (better) PPL at the smallest calibration sizes. Please clarify thresholds, metrics, and whether error bars or seeds alter this observation.

**3. Acceleration discussion is too thin.**
The efficiency section reports aggregate FLOPs and tokens/sec, but lacks kernel-level explanations of how sparse blocks and low-rank factors are scheduled or fused at inference time (e.g., whether sparse GEMV and low-rank ($UV^\top$) paths run in parallel, and whether gate/down projections can be overlapped). Detailing memory layout, operator choices, and fusion opportunities would strengthen the deployment story.

**4. Notation and terminology.**
Labels such as ***“RS-MoE100%high / RS-MoE80%high”*** are not self-explanatory in the main text. Please define them precisely (e.g., “percentage of budget allocated to sparse components”) where they first appear and in the figure/table captions.

**Questions:**

See weakness.

If these issues are addressed, I would be happy to raise my score.

**Details Of Ethics Concerns:**

NO or VERY MINOR ethics concerns only

---

> ### Author Response · Authors · 2025-11-26
> **Response to Reviewer BD9Z (Part 1/2)**
>
> We thank you for your thorough review. We appreciate the positive comments, as well as the thoughtful points raised. Please kindly find our response to your comments below. Additionally, all modifications to the manuscript have been highlighted in blue for easy reference. Please feel free to let us know if you have any additional concerns or questions.
>
> **Q1:** Limited novelty (composition of mature components). The sparse+low-rank paradigm is well established; activation-aware SVD, and sparse/low-rank fusion have prior art. The contribution reads more as a careful systemization on MoE than a fundamentally new algorithmic idea; the paper should delineate the novelty boundary more explicitly.
>
> **AQ1:**
> We appreciate the reviewer's candid assessment regarding component maturity. We acknowledge that low-rank approximation and pruning are established techniques. However, we respectfully clarify that RS-MoE is not merely a composition of these atomic operations. The core novelty lies in the Collaborative Decomposition framework designed specifically to address the structural coupling inherent to the SwiGLU architecture in modern MoEs.
>
> Generic approaches (like LoSparse or SoLA) typically treat the reconstruction of $W_{gate}$, $W_{up}$, and $W_{down}$ as independent optimization problems. In a SwiGLU expert where $y=\left(xW_{gate}\odot xW_{up}\right)W_{down}$, optimizing each matrix separately is suboptimal because it ignores how errors propagate through the multiplicative gating mechanism. RS-MoE treats the expert as a unified unit. Instead of minimizing local errors for each layer, we align their sparsity patterns and optimize the low-rank bases jointly using ridge regression on the coupled output to minimize the entire expert's error.
>
> To delineate the novelty boundary more explicitly, we summarize our key distinctions from generic sparse-low-rank methods below:
>
> | Feature | LoSparse (ICML'23) | SoLA (AAAI'25) | RS-MoE (Ours) |
> | :--- | :--- | :--- | :--- |
> | Optimization Scope | Independent (Matrix-wise) | Independent (Neuron-wise) | Collaborative (Coupled Triplet) |
> | Coupling Awareness | No (Additive) | No (Splits Neurons) | Yes (SwiGLU-Aware Unit) |
> | Compression Cost | Expensive Retraining | Training-Free | Training-Free |
> | Metric | Gradient Sensitivity | L2 Norm (Magnitude) | ACI (Peak + Downstream) |
> | Correction | Knowledge Distillation | None | Ridge Regression |
> | Sparsity Allocation | Uniform | Component-wise | MI-Guided |
>
> To quantifiably demonstrate that our collaborative design is a necessary architectural adaptation rather than a trivial composition, we compared RS-MoE against an independent baseline. This baseline simulates the naive application of sparse and low-rank strategies where each matrix is compressed separately.
>
> **Table 1: Collaborative vs. Independent Decomposition.**
> | Ratio | Method | Wiki. | PTB | C4 | ARC-e | HellaS. | Math. | Openb. | PIQA | WinoG. | Avg. |
> | :---: | :--- | :---: | :---: | :---: | :---: | :---: | :---: | :---: | :---: | :---: | :---: |
> | 0% | Original | 6.51 | 9.74 | 10.20 | 0.77 | 0.58 | 0.32 | 0.33 | 0.79 | 0.72 | 0.59 |
> | 20% | Independence | 7.17 | 11.13 | 12.03 | 0.74 | 0.54 | 0.31 | 0.33 | 0.77 | 0.70 | 0.57 |
> | | Collaboration | **6.74** | **10.42** | **11.28** | **0.76** | **0.56** | **0.32** | **0.33** | **0.77** | **0.71** | **0.58** |
> | 40% | Independence | 8.38 | 13.70 | 15.42 | 0.66 | 0.47 | 0.26 | 0.27 | 0.70 | 0.67 | 0.51 |
> | | Collaboration | **8.15** | **13.26** | **14.93** | **0.67** | **0.48** | **0.28** | **0.28** | **0.73** | **0.68** | **0.52** |
> | 60% | Independence | 10.97 | 19.36 | 23.91 | 0.58 | 0.39 | 0.24 | 0.21 | 0.66 | 0.64 | 0.45 |
> | | Collaboration | **9.95** | **18.29** | **22.52** | **0.59** | **0.40** | **0.26** | **0.26** | **0.68** | **0.65** | **0.47** |
>
> Table 1 above (Table 2 in the revised manuscript) reveals a clear distinction. Simply applying independent compression causes significant performance degradation as PPL rises from 9.95 to 10.97 at 60% sparsity. This result serves as concrete proof that the specific structural coupling is the key driver of performance
>
> **Action:** We have emphasized these distinctions in Section 2.2 and highlighted the ablation results in Section 4.3.

---

> ### Author Response · Authors · 2025-11-27
> **Response to Reviewer BD9Z (Part 2/2)**
>
> **Q2:** Figure 4 vs. text claim needs reconciliation. The paper states “RS-MoE is more robust when only a few calibration samples are provided,” yet in Figure 4 the plotted curves appear to show $D^2$-MoE achieving lower (better) PPL at the smallest calibration sizes. Please clarify thresholds, metrics, and whether error bars or seeds alter this observation.
>
> **AQ2:**
> We apologize for the confusion caused by a labeling error where the legends for $D^2$-MoE and RS-MoE were swapped in Figure 4. Once corrected, the visual data aligns perfectly with our text claim that RS-MoE consistently achieves lower PPL than $D^2$-MoE, particularly in the regime of few calibration samples.
>
> **Action:** We have updated Figure 4 in the revised manuscript to reflect this correction.
>
> **Q3:** Acceleration discussion is too thin. The efficiency section reports aggregate FLOPs and tokens/sec, but lacks kernel-level explanations of how sparse blocks and low-rank factors are scheduled or fused at inference time (e.g., whether sparse GEMV and low-rank ($UV^\top$) paths run in parallel, and whether gate/down projections can be overlapped). Detailing memory layout, operator choices, and fusion opportunities would strengthen the deployment story.
>
> **AQ3:**
> We agree that detailing the acceleration mechanism is crucial. Beyond the end-to-end latency speedups, we clarify our implementation strategy here. Instead of relying on inefficient unstructured sparse kernels, RS-MoE aggregates the preserved sparse components into compact dense blocks. Consequently, the acceleration is achieved by replacing large weight matrices with decomposed smaller matrices and leveraging existing hardware capabilities known as dense kernels. This avoids the latency and memory irregularity often seen with sparse operations. While our current implementation executes the sparse and low-rank paths sequentially, the significant reduction in FLOPs translates to real-world speedups.
>
> **Table 2: Online Inference Efficiency.**
> | Model | Ratio | Gate (ms) | Up (ms) | Down (ms) | Total (speedup) |
> | :--- | :---: | :---: | :---: | :---: | :---: |
> | **DeepSeekMoE-16B** | 0% | 0.69 | 0.69 | 0.73 | 2.11 |
> | | 20% | 0.65 | 0.65 | 0.69 | 1.99 (1.06x) |
> | | 40% | 0.47 | 0.47 | 0.49 | 1.43 (1.48x) |
> | | 60% | 0.36 | 0.37 | 0.41 | 1.14 (1.85x) |
> | **Mixtral-8x7B** | 0% | 13.35 | 14.09 | 16.55 | 43.99 |
> | | 20% | 10.96 | 10.88 | 11.35 | 33.19 (1.33x) |
> | | 40% | 8.09 | 8.12 | 9.75 | 25.96 (1.69x) |
> | | 60% | 5.65 | 5.81 | 6.21 | 17.67 (2.49x) |
> | **Qwen3-30B-A3B** | 0% | 0.39 | 0.40 | 0.43 | 1.22 |
> | | 20% | 0.40 | 0.40 | 0.44 | 1.24 (0.98x) |
> | | 40% | 0.34 | 0.36 | 0.37 | 1.07 (1.14x) |
> | | 60% | 0.21 | 0.21 | 0.23 | 0.65 (1.88x) |
>
> Table 2 (Table 2 in revised manuscript) confirms that RS-MoE delivers tangible speedups. For Mixtral-8x7B at 60% sparsity, we achieve a 2.49x speedup in matrix multiplication time. This demonstrates that RS-MoE is practical and delivers significant efficiency gains on standard hardware.
>
> **Action:** We have revised the efficiency analysis in Section 4.4 and added a detailed computational cost analysis in Appendix A.5.
>
> **Q4:** Notation and terminology. Labels such as “RS-MoE100%high / RS-MoE80%high” are not self-explanatory in the main text. Please define them precisely (e.g., “percentage of budget allocated to sparse components”) where they first appear and in the figure/table captions.
>
> **AQ4:**
> We thank the reviewer for identifying the ambiguous notation. We acknowledge that labels such as "RS-MoE100%high" were unclear. To address this, we have deprecated this terminology in our efficiency reporting. Instead, Table 6 in the revised manuscript (Table 3 below) now directly reports end-to-end latency in milliseconds alongside explicit speedup ratios. This eliminates the confusion caused by the previous notation and provides a standard view of inference performance.
>
> **Table 3: End-to-End Latency and Speedup Ratios.**
> | Ratio | Deepseek | Mixtral | Qwen |
> | :---: | :---: | :---: | :---: |
> | **0%** | 2.11 | 43.99 | 1.22 |
> | **20%** | 1.99 (1.06x) | 33.19 (1.33x) | 1.24 (0.98x) |
> | **40%** | 1.43 (1.48x) | 25.96 (1.69x) | 1.07 (1.14x) |
> | **60%** | 1.14 (1.85x) | 17.67 (2.49x) | 0.65 (1.88x) |
>
> **Action:** We have revised the efficiency analysis in Section 4.4 and updated relevant Table 6.

---

### Official Review · Reviewer_FPVr · 2025-10-30

**Soundness:** 3
**Presentation:** 3
**Contribution:** 2
**Rating:** 4
**Confidence:** 4

**Summary:**

The paper presents a well-motivated approach addressing MoE compression through collaborative decomposition that preserves expert functionality. The key insight that expert weights exhibit both sparse high-importance dimensions and low-rank structure after removing these dimensions is supported by empirical analysis (Figure 1; Sec. 3.1; p.2-3). Strong experimental results show RS-MoE achieves PPL 16.92 at 20% sparsity on DeepSeekMoE-16B-Base and 8.87 on Qwen3-30B-A3B, outperforming baselines (Tables 1-3; Sec. 4.2; p.7). However, theoretical justification for why the residual formulation improves optimization is largely intuitive without formal convergence analysis (Sec. 3.2; p.3-4). Some implementation details regarding BN placement and initialization sensitivity require clarification.

**Strengths:**

- **Clear problem identification and collaborative decomposition framework**
  - The manuscript documents activation peak distributions categorized into high/medium/low importance and demonstrates low-rank structure in whitened weights after removing high-importance components (Figure 1; Sec. 3.1; p.2), providing strong empirical motivation.
  - The collaborative unit concept couples corresponding dimensions of Wgate, Wup, and Wdown matrices (Sec. 3.2; p.4), ensuring functional integrity. The paper provides an equation that unfolds the expert computation, which clarifies this coupling.
  - The framework applies unified compression strategies to coupled dimensions (Sec. 3.2; p.3-4), mitigating spatial misalignment errors—important for preserving specialized knowledge.
- **Comprehensive importance scoring and decomposition strategy**
  - The ACI score integrates inner energy (mean, variance, peak activation) with downstream influence (alignment with subsequent layers) using a weighted combination (Algorithm 1; Sec. 3.3; p.5), providing robust importance quantification.
  - Activation-aware SVD projects weights into activation space for decomposition (Sec. 3.4; p.5-6), improving feature information retention.
  - Ridge regression constructs base weights to compensate for reconstruction errors (J(B) equation; Sec. 3.4; p.6), enhancing accuracy—critical for high compression ratios.
- **Strong experimental validation with thorough ablations**
  - Results show consistent improvements: PPL 16.92 vs. 18.19 (D2-MoE) at 60% sparsity on DeepSeekMoE; maintains 64% downstream accuracy at 20% sparsity on Qwen3 (Tables 1-3; Sec. 4.2-4.3; p.7-8), demonstrating state-of-the-art performance.
  - Ablations validate layer-wise sparsity allocation (Table 2), sparse vs. low-rank components (Table 3), and base weight construction methods (Table 4; Sec. 4.3; p.8), supporting design choices.
  - Robustness analysis with varying calibration samples shows superior stability compared to D2-MoE (Figure 4; Sec. 4.3; p.8), evidencing practical reliability.

**Weaknesses:**

- **Limited theoretical grounding for decomposition effectiveness**
  - The hypothesis that sparse-plus-low-rank decomposition preserves information is argued empirically; no formal analysis quantifies approximation error bounds or convergence guarantees (Sec. 3.1; p.2-3). This affects technical soundness and generalization confidence.
  - The ACI score combines multiple heuristics ($w\_{\text{mean}} = 0.4$, $w\_{\text{var}} = 0.05$, $w\_{\text{peak}} = 0.8$, $\gamma = 0.05$) without principled derivation or sensitivity analysis (Appendix A.2; p.13). Optimal hyperparameter selection lacks justification.
  - No theoretical characterization of when collaborative decomposition outperforms independent expert compression or conditions under which low-rank structure emerges after removing sparse components.
- **Incomplete ablation coverage and factor isolation**
  - Roles of batch normalization, initialization schemes, and learning-rate warm-up are acknowledged but not isolated via controlled experiments (Sec. 3.4; Sec. 4.2 warm-up note; p.4,7). This impacts experimental rigor and reproducibility.
  - Layer-wise parameter budget allocation uses quadratic programming with smoothness constraints but details are relegated to the appendix (Appendix A.3; p.13-14). The main text lacks a clear explanation of the budget determination process.
  - Analysis of activation sparsity across different layers/experts is limited to two examples (Figures 6-7; Appendix A.4; p.14-15). Broader empirical validation would strengthen generality claims.
- **Mathematical formulation clarity and notation consistency**
  - The collaborative unit formulation $Y = g \sum_{j} (\sigma(x \cdot W_{\text{gate},j,:}) \odot (x \cdot W_{\text{up},j,:})) W_{\text{down},:,j}$ (Sec. 3.2; p.4) could benefit from explicit dimensionality annotations and a clearer distinction between element-wise and matrix operations.
  - The Ridge regression objective $J(B) = \|Y - H_c B^T\| + \lambda \|B\|$ uses an unspecified norm, and the regularization parameter λ=1e-3 lacks justification (Sec. 3.4; Appendix A.2; p.6,13).
  - Response magnitude statistics require explicit definitions for computation points (post-BN/pre-ReLU) and aggregation procedures (Sec. 3.4; p.5). Current descriptions are informal.
- **Reproducibility and resource reporting gaps**
  - Implementation uses 128 Wikitext2 samples truncated to 2048 tokens on NVIDIA A800 GPUs (Sec. 4.1; Appendix A.2; p.7,13), but memory footprints, training wall-clock times, and exact hardware specs per model/depth are not reported, limiting adoption planning.
  - Efficiency analysis (Table 5) reports FLOPs and throughput for only 60% sparsity on Mixtral (Sec. 4.4; p.8). Comprehensive efficiency metrics across models and compression ratios would strengthen practical utility claims.

**Questions:**

- **Strengthen theoretical framework**
  - Provide formal approximation error bounds for the sparse-plus-low-rank decomposition, showing the reconstruction error $\epsilon$ as a function of sparsity ratio $s$ and rank $r$. Include proof sketches in the main text with full derivations in the appendix.
  - Conduct a sensitivity analysis for the ACI hyperparameters $(w\_{\text{mean}}, w\_{\text{var}}, w\_{\text{peak}}, \gamma)$ across different models and datasets. Provide guidelines or adaptive selection procedures based on model characteristics.
  - Develop theoretical conditions that characterize when collaborative decomposition outperforms independent compression, possibly via an analysis of weight coupling through intermediate activations.
- **Expand ablation studies**
  - Execute systematic ablations isolating batch normalization (BN-on vs. BN-off), initialization schemes (Xavier vs. He vs. default), and learning-rate warm-up across multiple depths and models (analogous to Sec. 3.4; Sec. 4.2; p.4,7).
  - Provide layer-wise analyses for projection shortcut options (A/B/C) showing gradient flow or activation statistics to explain where projections materially impact performance (extending Table 3 analysis; p.6).
  - Conduct necessity/sufficiency tests: plain nets with BN, identity shortcuts selectively removed, and alternative decomposition strategies to validate the attribution of gains to the collaborative framework.
- **Clarify mathematical formulations**
  - Add explicit dimensionality annotations to all equations (e.g., specify $x \in \mathbb{R}^{1 \times d}$, $W\_{\text{gate},j,:} \in \mathbb{R}^{1 \times d}$, output dimensions). Include operator precedence diagrams for the residual block activation order (enhancing Fig. 2; Sec. 3.2; p.3).
  - Specify the norm types in the ridge regression objective ($L\_2$ is assumed but not stated) and provide principled selection criteria for $\lambda$ based on model scale or data characteristics (Sec. 3.4; p.6).
  - Formalize the response magnitude statistics: define computation points, aggregation across layers/batches, and report summary tables with confidence intervals (clarifying Fig. 7; Sec. 4.2; p.8).
- **Enhance reproducibility reporting**
  - Report memory usage (peak GPU memory, parameter storage), training time (wall-clock per epoch/iteration), and detailed hardware specifications for each model/depth configuration (supplementing Table 1; Sec. 3.4; p.4-6).
  - Expand the efficiency analysis (Table 5) to cover all compression ratios (20%, 40%, 60%) across all models with metrics including FLOPs, throughput, latency, and memory footprint.

---

> ### Author Response · Authors · 2025-11-26
> **Response to Reviewer FPVr (Part 1/2)**
>
> We thank you for your thorough review. We appreciate the positive comments, as well as the thoughtful points raised. Please kindly find our response to your comments below. Additionally, all modifications to the manuscript have been highlighted in blue for easy reference. Please feel free to let us know if you have any additional concerns or questions.
>
> >**Q1.1 & W1.1:** Provide formal approximation error bounds for the sparse-plus-low-rank decomposition, showing the reconstruction error $\epsilon$ as a function of sparsity ratio $s$ and rank $r$. Include proof sketches in the main text with full derivations in the appendix.
>
> **AQ1.1 & AW1.1:**
> We appreciate this insightful suggestion regarding theoretical guarantees. We acknowledge that deriving strict error bounds for MoE compression is a valuable direction. However, we clarify that establishing such bounds remains an open challenge due to the discrete, non-differentiable nature of the MoE routing mechanism and the non-linearity of the SwiGLU activation.
>
> To serve as a practical verification in lieu of closed-form bounds, we focused on strong empirical validation. Table 2 below (Table 2 in the revised manuscript) shows our collaborative design consistently outperforms independent decomposition, effectively minimizing reconstruction error $\epsilon$. Furthermore, our sensitivity analysis in Figure 10 confirms the stability of our approximation across a wide range of hyperparameters.
>
> >**Q1.2 & W1.2:** Conduct a sensitivity analysis for the ACI hyperparameters $\boldsymbol{(w_{\mathrm{mean}},w_{\mathrm{var}},w_{\mathrm{peak}},\gamma)}$ across different models and datasets. Provide guidelines or adaptive selection procedures based on model characteristics.
>
> **AQ1.2 & AW1.2:**
> We agree that robustness analysis is essential. In our analysis, we identified $w_{peak}$ as the most critical hyperparameter governing the selection of sparse components.
>
> To validate its stability, we conducted the requested sensitivity analysis. Figure 10 (Appendix A.6) reveals a clear "robustness zone" where performance remains stable even as $w_{peak}$ varies between 0.8 and 1.2. Additionally, we performed a component ablation study (Table 9 in the revised manuscript) to justify our default configuration:
>
> **Table 1: ACI Hyperparameter Ablation.**
> | Method Variant | Inner Stats ($w_{peak}$) | Downstream ($\gamma$) | PPL (Wikitext2) |
> | :--- | :---: | :---: | :---: |
> | **Full ACI (Ours)** | ✓ | ✓ | **9.95** |
> | w/o Downstream | ✓ | X | 10.12 |
> | w/o Peak (Mean+Var only) | X | ✓ | 15.76 |
> | Weight Magnitude | X | X | 21.35 |
>
> The results confirm that while activation peaks are dominant, the downstream constraint $\gamma$ provides a necessary refinement for optimal performance.
>
> **Action:** We have included this sensitivity analysis in Appendix A.6.
>
> >**Q1.3 & W1.3:** Develop theoretical conditions that characterize when collaborative decomposition outperforms independent compression, possibly via an analysis of weight coupling through intermediate activations.
>
> **AQ1.3 & AW1.3:**
> We thank the reviewer for this interesting theoretical angle. We clarify that the core condition for superiority lies in the structural coupling of the SwiGLU architecture. The expert computation $Y = \sum_{j} \left( \sigma(x \cdot W_{gate, j:}) \odot (x \cdot W_{up, j:}) \right) W_{down, :j}$ establishes a strict structural dependency.
>
> Empirically, this hypothesis is confirmed by the consistent performance gap shown in our experiments:
>
> **Table 2: Collaborative vs. Independent Decomposition.**
> | Ratio | Method | Wiki. | PTB | C4 | ARC-e | HellaS. | Math. | Openb. | PIQA | WinoG. | Avg. |
> | :---: | :--- | :---: | :---: | :---: | :---: | :---: | :---: | :---: | :---: | :---: | :---: |
> | 0% | Original | 6.51 | 9.74 | 10.20 | 0.77 | 0.58 | 0.32 | 0.33 | 0.79 | 0.72 | 0.59 |
> | 20% | Independence | 7.17 | 11.13 | 12.03 | 0.74 | 0.54 | 0.31 | 0.33 | 0.77 | 0.70 | 0.57 |
> | | Collaboration | **6.74** | **10.42** | **11.28** | **0.76** | **0.56** | **0.32** | **0.33** | **0.77** | **0.71** | **0.58** |
> | 40% | Independence | 8.38 | 13.70 | 15.42 | 0.66 | 0.47 | 0.26 | 0.27 | 0.70 | 0.67 | 0.51 |
> | | Collaboration | **8.15** | **13.26** | **14.93** | **0.67** | **0.48** | **0.28** | **0.28** | **0.73** | **0.68** | **0.52** |
> | 60% | Independence | 10.97 | 19.36 | 23.91 | 0.58 | 0.39 | 0.24 | 0.21 | 0.66 | 0.64 | 0.45 |
> | | Collaboration | **9.95** | **18.29** | **22.52** | **0.59** | **0.40** | **0.26** | **0.26** | **0.68** | **0.65** | **0.47** |
>
> As detailed in Table 2 (Table 2 in the revised manuscript), the Collaborative approach maintains a PPL advantage of approximately 1.0 over the Independent baseline at 60% sparsity. This gap quantifies the benefit of preserving the coupled structure.
>
> **Action:** We have added this empirical analysis in Section 4.3 and Table 2 (Table 2 in revised manuscript).

---

> ### Author Response · Authors · 2025-11-26
> **Response to Reviewer FPVr (Part 2/2)**
>
> >**Q2 & W2:** Could you execute ablations on batch normalization, initialization, warm-up, and projection shortcuts (A/B/C) to validate necessity?
>
> **AQ2 & AW2:**
> We appreciate the suggestion to inspect these training components. We clarify that RS-MoE is a training-free post-processing framework applied directly to pre-trained Transformers. Unlike pruning-aware training methods, our approach relies solely on analytical solutions (SVD and Ridge Regression) and does not involve gradient-based optimization loops. Consequently, hyperparameters such as batch normalization schedules, initialization schemes, or learning rate warm-ups are not applicable to our pipeline.
>
> >**Q3 & W3:** Could you clarify mathematical details regarding dimensionality annotations, the Ridge Regression objective, and response magnitude statistics?
>
> **AQ3 & AW3:**
> We have strengthened the mathematical formulation following your advice. Specifically:
> - Dimensionality: We updated Section 3.2 to define dimensionality explicitly ($x \in \mathbb{R}^{1 \times n}$).
> - Ridge Regression: We clarified in Section 3.4 that the objective minimizes the Frobenius norm ($||\cdot||_F^2$) of the reconstruction error.
> - Response Magnitude: We formalized the definition in Appendix A.4 by specifying that the statistics are computed on the intermediate expert activations. We also clarified that these values are aggregated across the token sequence dimension of the calibration set to determine the ACI scores.
>
> **Action:** We have updated the relevant Section 3.2, Section 3.4, and Appendix A.4 with these clarifications.
>
> >**Q4 & W4:** Could you enhance reproducibility by reporting detailed resource usage and expanding the efficiency analysis?
>
> **AQ4 & AW4:**
> We fully agree on the importance of transparency regarding resource consumption. We have expanded our analysis to cover both the offline compression cost and online inference speedups.
>
> **Table 3: Offline Compression Cost.**
> | Stage | Metric | DeepSeekMoE-16B | Mixtral-8x7B | Qwen3-30B-A3B |
> | :--- | :--- | :---: | :---: | :---: |
> | **MINE** | Time Cost | 2 mins | 5 mins | 4 mins |
> | | Peak VRAM | 35.72 GB | 122.47 GB | 72.02 GB |
> | **ACI** | Time Cost | 6 mins | 9 mins | 8 mins |
> | | Peak VRAM | 34.86 GB | 121.56 GB | 71.37 GB |
> | **Slice & SVD** | Time Cost | 24 mins | 61 mins | 43 mins |
> | | Peak VRAM | 42.82 GB | 125.67 GB | 71.58 GB |
>
> **Table 4: Online Inference Efficiency.**
> | Model | Ratio | Gate (ms) | Up (ms) | Down (ms) | Total (speedup) |
> | :--- | :---: | :---: | :---: | :---: | :---: |
> | **DeepSeekMoE-16B** | 0% | 0.69 | 0.69 | 0.73 | 2.11 |
> | | 20% | 0.65 | 0.65 | 0.69 | 1.99 (1.06x) |
> | | 40% | 0.47 | 0.47 | 0.49 | 1.43 (1.48x) |
> | | 60% | 0.36 | 0.37 | 0.41 | 1.14 (1.85x) |
> | **Mixtral-8x7B** | 0% | 13.35 | 14.09 | 16.55 | 43.99 |
> | | 20% | 10.96 | 10.88 | 11.35 | 33.19 (1.33x) |
> | | 40% | 8.09 | 8.12 | 9.75 | 25.96 (1.69x) |
> | | 60% | 5.65 | 5.81 | 6.21 | 17.67 (2.49x) |
> | **Qwen3-30B-A3B** | 0% | 0.39 | 0.40 | 0.43 | 1.22 |
> | | 20% | 0.40 | 0.40 | 0.44 | 1.24 (0.98x) |
> | | 40% | 0.34 | 0.36 | 0.37 | 1.07 (1.14x) |
> | | 60% | 0.21 | 0.21 | 0.23 | 0.65 (1.88x) |
>
> As shown in Table 3 (Table 7 in revised manuscript), the offline compression is highly efficient, taking only 2 to 5 minutes for the MINE estimation stage. Furthermore, Table 4 (Table 8 in revised manuscript) confirms the practical benefits of our method, demonstrating a speedup of 1.88x to 2.49x at 60% sparsity across different models.
>
> **Action:** We have revised the content of Section 4.4 Efficiency Analysis and added a detailed computational cost analysis in Appendix A.5.

---

### Official Review · Reviewer_YZg1 · 2025-11-01

**Soundness:** 3
**Presentation:** 2
**Contribution:** 3
**Rating:** 6
**Confidence:** 3

**Summary:**

This paper introduces **RS-MoE**, a post-training compression framework tailored to Mixture-of-Experts (MoE) LLMs. The core idea is to jointly exploit MoE structure inside each expert by coupling three matrices around the SwiGLU intermediate (the “collaborative unit”): the rows of \(W_{\text{up}}\) and \(W_{\text{gate}}\) and the corresponding column of \(W_{\text{down}}\). For each index \(j\), RS-MoE assigns high-importance dimensions to a sparse branch and compresses the rest via a low-rank branch, then uses a small ridge-regression fit to reduce residual error. Dimensional importance is estimated by ACI (Anomalous Contribution Integration), combining routed activation statistics (mean/variance/peaks) with a proxy for downstream influence. Finally, RS-MoE performs mutual-information–guided layer allocation, using an MI estimator to distribute sparsity non-uniformly across depth. Experiments on several MoE models (e.g., Mixtral, DeepSeek-MoE, Qwen-MoE) show that RS-MoE improves perplexity/zero-shot accuracy versus expert-level pruning/merging baselines and reports some throughput/FLOPs trade-offs under certain settings.

**Strengths:**

- **MoE-aware design.** The collaborative-unit view ties parameters that co-determine each expert dimension, encouraging **aligned** sparse/low-rank decisions and helping preserve expert specialization.
- **Importance beyond magnitude.** ACI uses routed activation statistics and a downstream proxy to rank dimensions, typically outperforming magnitude-only criteria in retention quality.
- **Complementary decompositions.** High-importance dimensions remain sparse; the remainder uses activation-aware SVD, with **ridge** to absorb truncation errors—often superior to pure pruning or pure low-rank.
- **Non-uniform depth allocation.** MI-guided scheduling assigns sparsity where layers are more redundant, beating uniform or naive schedules in most cases.
- **Small calibration footprint.** Uses a modest number of calibration samples to collect activations/routing, showing reasonable stability to sample count.

**Weaknesses:**

1. **MoE-specific leverage vs. “applying old tricks.”** It is not yet clear what intrinsically MoE-specific insight drives the gains beyond standard sparse/low-rank compression. The paper claims index-aligned coupling across \(W_{\text{up}}, W_{\text{gate}}, W_{\text{down}}\), but the causal link to preserving expert specialization (vs. independent per-matrix pruning) is insufficiently isolated. Without ablations that remove/disable the MoE coupling, the method risks looking like a generic compressor “ported” to MoE.

2. **Runtime benefits are not convincingly established.** Storage/compression metrics and FLOPs proxies are provided, but **end-to-end speedups**  are limited or model-specific. Without comprehensive latency/tokens-per-second results across multiple MoE models and sparsity settings, the efficiency claim remains tentative.

3. **Ablation granularity and causal attribution.** The contributions of (i) MoE index coupling, (ii) the importance metric’s components, (iii) low-rank rank selection, and (iv) MI-guided layer allocation are not disentangled. It is hard to tell which component actually matters, and whether the MoE-specific parts—rather than generic sparse/low-rank—drive the gains.

**Questions:**

1. **ACI ablations:** Break down the contributions of inner-activation statistics vs. downstream-influence components. How sensitive is performance to the weighting of these terms?

2. **MI estimator cost/stability:** What is the training time and variance of the MI estimator across runs? Is there a simpler heuristic (e.g., depth-based) that approximates MI enough to avoid training it?

---

> ### Author Response · Authors · 2025-11-26
> **Response to Reviewer YZg1 (Part 1/3)**
>
> We thank you for your thorough review. We appreciate the positive comments, as well as the thoughtful points raised. Please kindly find our response to your comments below. Additionally, all modifications to the manuscript have been highlighted in blue for easy reference. Please feel free to let us know if you have any additional concerns or questions.
>
> >**Q1:** ACI ablations: Break down the contributions of inner-activation statistics vs. downstream-influence components. How sensitive is performance to the weighting of these terms?
> >**W3.2:** Ablation granularity and causal attribution. The contributions of the importance metric’s components are not disentangled.
>
> **AQ1 & AW3.2:**
> We appreciate this suggestion. Isolating the contributions of our Anomalous Contribution Integration (ACI) metric components is crucial. Our new ablation study (Table 1, corresponding to Table 9 in the revised manuscript) empirically confirms that while the inner-activation statistics $w_{peak}$ are the dominant source of specialized knowledge, the downstream-influence $\gamma$ component provides a critical, non-trivial boost to overall performance:
>
> **Table 1: ACI Ablation Study.**
> | Method Variant | Inner Stats ($w_{peak}$) | Downstream ($\gamma$) | PPL (Wikitext2) |
> | :--- | :---: | :---: | :---: |
> | **Full ACI (Ours)** | ✓ | ✓ | **9.95** |
> | w/o Downstream | ✓ | X | 10.12 |
> | w/o Peak (Mean+Var only) | X | ✓ | 15.76 |
> | Weight Magnitude | X | X | 21.35 |
>
> The results reveal two critical insights. First, removing the Inner Statistics $w_{peak}$ causes a sharp performance drop to 15.76. This confirms that activation peaks are the dominant factor for identifying specialized knowledge. Second, incorporating the Downstream Influence ($\gamma$) further reduces the PPL from 10.12 to 9.95. This proves that considering the alignment with the next layer provides a non-trivial boost to the overall integrity of the expert network.
>
> Furthermore, we verified the robustness of our default setting ($w_{peak}=0.8$) through a sensitivity analysis shown in Figure 10 of the revision. This analysis confirms that the PPL remains in a highly stable robustness zone where $\Delta$ PPL $< 0.5$ even when $w_{peak}$ fluctuates between 0.8 and 1.2. This empirical evidence supports the stability of our proposed ACI mechanism.
>
> **Action:** We have added the detailed ablation study in Table 9 and the sensitivity analysis in Figure 10 to Appendix A.6.
>
> >**Q2:** MI estimator cost/stability: What is the training time and variance of the MI estimator across runs? Is there a simpler heuristic (e.g., depth-based) that approximates MI enough to avoid training it?
> >**W3.4:** Ablation granularity and causal attribution. The contributions of the MI-guided layer allocation are not disentangled.
>
> **AQ2 & W3.4:**
> We appreciate the practical concern regarding computational overhead. We clarify that the MINE estimator incurs a negligible one-off cost while providing significant performance gains over simple heuristics.
>
> As detailed in Table 2 (Table 7 in the revised manuscript), the estimation takes only 2 to 5 minutes per model with stable convergence.
>
> **Table 2: MINE Training Cost.**
> | Metric | DeepSeekMoE-16B | Mixtral-8x7B | Qwen3-30B-A3B |
> | :--- | :---: | :---: | :---: |
> | Time Cost | 2 mins | 5 mins | 4 mins |
> | Peak VRAM | 35.72 GB | 122.47 GB | 72.02 GB |
>
> Regarding simpler heuristics, we refer the reviewer to Table 3 of our original manuscript (reproduced below), where we explicitly compared our MI-guided strategy against Uniform and Reverse (depth-based) allocations. The results show that our MI-based allocation achieves a PPL of 12.11, significantly outperforming uniform and reverse strategies. This confirms that simple heuristics fail to capture the complex redundancy patterns of MoE models, making the minimal time investment in MI estimation essential for optimal performance.
>
> **Table 3: Ablation of Sparsity Allocation Strategies.**
> | Strategy | Wikitext-2 | PTB | C4 | Average |
> | :--- | :---: | :---: | :---: | :---: |
> | Uniform | 8.12 | 13.59 | 15.20 | 12.30 |
> | Reverse | **8.10** | 13.94 | 15.46 | 12.50 |
> | RS-MoE | 8.15 | **13.26** | **14.93** | **12.11** |
>
> **Action:** We have detailed the offline computation cost of MI estimator in Table 7 of Appendix A.5.

---

> ### Author Response · Authors · 2025-11-27
> **Response to Reviewer YZg1 (Part 2/3)**
>
> >**W1:** MoE-specific leverage vs. “applying old tricks.”  It is not yet clear what intrinsically MoE-specific insight drives the gains beyond standard sparse/low-rank compression. The paper claims index-aligned coupling across ($W_{\text{up}}, W_{\text{gate}}, W_{\text{down}}$), but the causal link to preserving expert specialization (vs. independent per-matrix pruning) is insufficiently isolated.
> >**W3.1:** Ablation granularity and causal attribution. The contributions of the MoE index coupling are not disentangled.
>
> **AW1 & AW3.1:**
> We absolutely agree that isolating the benefits of our MoE-specific Collaborative Decomposition is mandatory. We clarify that the superiority of RS-MoE stems from the "structural coupling" of the SwiGLU matrices rather than merely applying Low-Rank and Sparse approximations. Treating $W_{up}$, $W_{gate}$, $W_{down}$ as independent entities ignores the intrinsic mathematical link within the expert, which leads to error propagation.
>
> To empirically verify this, we compared our Collaborative strategy against an "Independent" baseline where the same compression budget is applied to each matrix separately.
>
> **Table 3: Collaborative vs. Independent Decomposition.**
> | Ratio | Method | Wiki. | PTB | C4 | ARC-e | HellaS. | Math. | Openb. | PIQA | WinoG. | Avg. |
> | :---: | :--- | :---: | :---: | :---: | :---: | :---: | :---: | :---: | :---: | :---: | :---: |
> | 0% | Original | 6.51 | 9.74 | 10.20 | 0.77 | 0.58 | 0.32 | 0.33 | 0.79 | 0.72 | 0.59 |
> | 20% | Independence | 7.17 | 11.13 | 12.03 | 0.74 | 0.54 | 0.31 | 0.33 | 0.77 | 0.70 | 0.57 |
> | | Collaboration | **6.74** | **10.42** | **11.28** | **0.76** | **0.56** | **0.32** | **0.33** | **0.77** | **0.71** | **0.58** |
> | 40% | Independence | 8.38 | 13.70 | 15.42 | 0.66 | 0.47 | 0.26 | 0.27 | 0.70 | 0.67 | 0.51 |
> | | Collaboration | **8.15** | **13.26** | **14.93** | **0.67** | **0.48** | **0.28** | **0.28** | **0.73** | **0.68** | **0.52** |
> | 60% | Independence | 10.97 | 19.36 | 23.91 | 0.58 | 0.39 | 0.24 | 0.21 | 0.66 | 0.64 | 0.45 |
> | | Collaboration | **9.95** | **18.29** | **22.52** | **0.59** | **0.40** | **0.26** | **0.26** | **0.68** | **0.65** | **0.47** |
>
> As shown in the table above (Table 2 in revised manuscript), the difference becomes pronounced at high compression rates. At a 60% ratio, the Collaborative approach outperforms the Independent baseline by a significant margin of roughly 1.0 PPL. This quantifiably proves that our design effectively preserves the functional integrity of the expert units by minimizing the alignment error across the three matrices.
>
> **Action:** We added the detailed ablation study in Table 2 in revised manuscript to delineate the novelty boundary.
>
> >**W2:** Runtime benefits are not convincingly established. Storage/compression metrics and FLOPs proxies are provided, but end-to-end speedups are limited or model-specific.
>
> **AW2:**
> We appreciate the reviewer's suggestion on efficiency. To substantiate our efficiency claims beyond FLOPs proxies, we have conducted comprehensive end-to-end latency measurements for matrix multiplications on NVIDIA A800 GPUs.
>
> **Table 4: Online Inference Efficiency.**
> | Model | Ratio | Gate (ms) | Up (ms) | Down (ms) | Total (speedup) |
> | :--- | :---: | :---: | :---: | :---: | :---: |
> | **DeepSeekMoE-16B** | 0% | 0.69 | 0.69 | 0.73 | 2.11 |
> | | 20% | 0.65 | 0.65 | 0.69 | 1.99 (1.06x) |
> | | 40% | 0.47 | 0.47 | 0.49 | 1.43 (1.48x) |
> | | 60% | 0.36 | 0.37 | 0.41 | 1.14 (1.85x) |
> | **Mixtral-8x7B** | 0% | 13.35 | 14.09 | 16.55 | 43.99 |
> | | 20% | 10.96 | 10.88 | 11.35 | 33.19 (1.33x) |
> | | 40% | 8.09 | 8.12 | 9.75 | 25.96 (1.69x) |
> | | 60% | 5.65 | 5.81 | 6.21 | 17.67 (2.49x) |
> | **Qwen3-30B-A3B** | 0% | 0.39 | 0.40 | 0.43 | 1.22 |
> | | 20% | 0.40 | 0.40 | 0.44 | 1.24 (0.98x) |
> | | 40% | 0.34 | 0.36 | 0.37 | 1.07 (1.14x) |
> | | 60% | 0.21 | 0.21 | 0.23 | 0.65 (1.88x) |
>
> Table 3 (Table 8 in the revised manuscript) provides concrete evidence of these efficiency gains. Take Mixtral-8x7B as a prime example: we achieve a significant 2.49x acceleration at 60% sparsity. This speedup is driven by our design choice to replace large weight matrices with decomposed, smaller dense matrices. By leveraging hardware-friendly dense kernels, we effectively bypass the latency overheads that typically plague sparse matrix operations.
>
> **Action:** We have included these latency results in Table 8 of Appendix A.5 to substantiate our efficiency claims.

---

> > ### Author Response · Authors · 2025-12-01
> > **Response to Reviewer YZg1 (Part 3/3)**
> >
> > > **W3.3:** Ablation granularity and causal attribution. The contributions of the low-rank rank selection are not disentangled.
> >
> > **AW3.3:**
> > Thanks for this suggestion. We further address the specific contribution of our low-rank rank selection strategy by isolating its effect in the newly added Table 5 (Table 11 in the revised manuscript). We compared our energy-based adaptive allocation against a standard fixed-rank SVD baseline where the rank is uniformly distributed across modules.
> >
> > **Table 5: Ablation of Low-Rank Rank Selection Strategies.**
> > | Rank Allocation Strategy | PPL |
> > | :--- | :---: |
> > | Standard SVD (Fixed Rank) | 17.03 |
> > | **RS-MoE (Energy-based Adaptive)** | **16.92** |
> >
> > The table above confirms that our adaptive approach outperforms the fixed-rank baseline with a PPL of 16.92 versus 17.03. This gain stems from prioritizing matrices with slower spectral decay. By dynamically allocating the budget based on energy retention, we effectively preserve fidelity in information-dense components.
> >
> > **Action:** We have added Appendix A.9 which details the mathematical formulation of our energy-based allocation strategy and includes the ablation results in Table 11.

---

### Official Review · Reviewer_DdvH · 2025-11-01

**Soundness:** 2
**Presentation:** 2
**Contribution:** 1
**Rating:** 2
**Confidence:** 4

**Summary:**

This paper presents a novel MoE model decomposition method, where model weights are treated in a more fine-grained way. It first identifies model weights that can produce high activation values. The important part is preserved and only the less important part is being decomposed. The proposed method also use mutual information for allocating compression ratios in different layers. Extensive experiments are carried out to demonstrate the superiority of the proposed method.

**Strengths:**

1. This paper is well-written and easy to follow.

2. The experiments carried out in this paper are extensive and comprehensive, which involves multiple common Moe models under different compression ratios. But the evaluation part is also limited, details refer to Weaknesses below.

**Weaknesses:**

1. Lack of novelty and also discussion about two important previous works. The key findings presented in this paper are originally comes from *LoSparse: Structured Compression of Large Language Models based on Low-Rank and Sparse Approximation* (ICML'23) and *SoLA: Leveraging Soft Activation Sparsity and Low-Rank Decomposition for Large Language Model Compression* (AAAI'25). Authors seem missing these two works, and should clarify the novelty compared with these two papers (except for adapting to MoE models).

2. Limited evaluation. The experiments involved in this paper don't include any generative tasks. Authors are suggested to add more experiments about generative tasks, such as arithmetic reasoning and summarization.

3. Need efficiency improvement evaluation. The proposed method might reproduce too many small matrix and thus cause significant performance degradation during the generation. Authors need to evaluate the generation performance to demonstrate it won't incur severe performance issues.

**Questions:**

See Weaknesses above.

---

> ### Author Response · Authors · 2025-11-26
> **Response to Reviewer DdvH (Part 1/2)**
>
> We thank the reviewer for the constructive feedback. We address the specific concerns below.
>
> >**W1:** Lack of novelty and also discussion about two important previous works.
>
> **A1:**
> We thank the reviewer for highlighting LoSparse and SoLA. We fully agree that these are essential precursors that established the potential of combining sparse and low-rank approximations. However, we respectfully clarify that RS-MoE is not merely a direct application of these methods to MoE models. Our primary novelty lies in addressing the structural coupling inherent to the SwiGLU architecture in modern MoEs, which prior independent decomposition methods do not address.
>
> Conceptually, methods like SoLA operate by identifying prime neurons and decomposing the remaining weights. While this is effective for LLaMA models, it generally treats the reconstruction of $W_{gate}$, $W_{up}$, $W_{down}$ as independent optimization problems. In a SwiGLU expert where $y=\left(xW_{gate}\odot xW_{up}\right)W_{down}$, optimizing the reconstruction error of each matrix separately is suboptimal because it ignores how errors propagate through the multiplicative gating mechanism. RS-MoE treats the expert as a unified unit. Instead of minimizing local errors for each layer, we align their sparsity patterns and optimize the low-rank bases jointly using ridge regression on the coupled output to minimize the entire expert's error.
>
> Our key distinctions are summarized below:
>
> **Table 1: Distinction from LoSparse and SoLA.**
> | Feature | LoSparse (ICML'23) | SoLA (AAAI'25) | RS-MoE (Ours) |
> | :--- | :--- | :--- | :--- |
> | Optimization Scope | Independent (Matrix-wise) | Independent (Neuron-wise) | Collaborative (Coupled Triplet) |
> | Coupling Awareness | No (Additive) | No (Splits Neurons) | Yes (SwiGLU-Aware Unit) |
> | Compression Cost | Expensive Retraining | Training-Free | Training-Free |
> | Metric | Gradient Sensitivity | L2 Norm (Magnitude) | ACI (Peak + Downstream) |
> | Correction | Knowledge Distillation | None | Ridge Regression |
> | Sparsity Allocation | Uniform | Component-wise | MI-Guided Optimal |
>
> To prove that our collaborative design is essential rather than a trivial adaptation, we compared RS-MoE against an independent baseline. This baseline simulates the application of SoLA and LoSparse strategies, where each matrix is compressed separately. Table 2 demonstrates a main difference:
>
> **Table 2: Collaborative vs. Independent Decomposition.**
> | Ratio | Method | Wiki. | PTB | C4 | ARC-e | HellaS. | Math. | Openb. | PIQA | WinoG. | Avg. |
> | :---: | :--- | :---: | :---: | :---: | :---: | :---: | :---: | :---: | :---: | :---: | :---: |
> | 0% | Original | 6.51 | 9.74 | 10.20 | 0.77 | 0.58 | 0.32 | 0.33 | 0.79 | 0.72 | 0.59 |
> | 20% | Independence | 7.17 | 11.13 | 12.03 | 0.74 | 0.54 | 0.31 | 0.33 | 0.77 | 0.70 | 0.57 |
> | | Collaboration | **6.74** | **10.42** | **11.28** | **0.76** | **0.56** | **0.32** | **0.33** | **0.77** | **0.71** | **0.58** |
> | 40% | Independence | 8.38 | 13.70 | 15.42 | 0.66 | 0.47 | 0.26 | 0.27 | 0.70 | 0.67 | 0.51 |
> | | Collaboration | **8.15** | **13.26** | **14.93** | **0.67** | **0.48** | **0.28** | **0.28** | **0.73** | **0.68** | **0.52** |
> | 60% | Independence | 10.97 | 19.36 | 23.91 | 0.58 | 0.39 | 0.24 | 0.21 | 0.66 | 0.64 | 0.45 |
> | | Collaboration | **9.95** | **18.29** | **22.52** | **0.59** | **0.40** | **0.26** | **0.26** | **0.68** | **0.65** | **0.47** |
>
> Removing the collaborative mechanism leads to a significant performance drop. For instance, PPL increases from 9.95 to 10.97 on WikiText2 at 60% sparsity. It proves that collaborative decomposition is the key factor enabling high compression rates and distinguishes our work from prior approaches.
>
> Furthermore, unlike SoLA, which relies on activation norms suited for dense models, our ACI metric explicitly captures activation peaks and downstream influence. This design is grounded in recent findings on MoE interpretability $^{[1]}$, which identify activation peaks rather than simple magnitude as the primary indicator of expert specialization. As shown in Table 3 (Table 9 in the revised manuscript), relying solely on basic statistics such as mean and variance degrades PPL to 15.76. Incorporating peak and downstream components recovers PPL to 9.95, demonstrating the superiority of our metric over conventional importance measures.
>
> **Table 3: Efficiency of ACI.**
> | Method Variant | Inner Stats ($w_{peak}$) | Downstream ($\gamma$) | PPL (Wikitext2) |
> | :--- | :---: | :---: | :---: |
> | **Full ACI (Ours)** | ✓ | ✓ | **9.95** |
> | w/o Downstream | ✓ | X | 10.12 |
> | w/o Peak (Mean+Var only) | X | ✓ | 15.76 |
> | Weight Magnitude | X | X | 21.35 |
>
> **Action:** We have revised Section 2.2 to include and distinguish RS-MoE from these relevant works explicitly, and added ablation tables to Section 4.3 and Appendix A.6 for clarity.
>
> [1] Su, Zunhai, et al. "Unveiling super experts in mixture-of-experts large language models." arXiv preprint arXiv:2507.23279 (2025).

---

> ### Author Response · Authors · 2025-11-27
> **Response to Reviewer DdvH (Part 2/2)**
>
> >**W2:** Limited evaluation. The experiments involved in this paper don't include any generative tasks. Authors are suggested to add more experiments about generative tasks, such as arithmetic reasoning and summarization.
>
> **A2:**
> We agree that evaluation on generative tasks is essential for validating the preservation of model fidelity.
>
> Regarding arithmetic reasoning, Table 1 in our original manuscript evaluates RS-MoE on MathQA. The results show that RS-MoE maintains competitive reasoning ability, maintaining 0.28 accuracy versus the 0.32 baseline at 40% sparsity. It consistently outperforms other compression methods in this domain.
>
> To further address the request for summarization and long-form generation, we have conducted new experiments on the CNN/DailyMail dataset, evaluated using ROUGE scores. Table 4 (Table 10 in the revised manuscript) summarizes the results:
>
> **Table 4: Generative Task Performance on DeepseekMoE-16B.**
> | Method        | Ratio | R-1   | R-2  | R-L   |
> |---------------|-------|-------|------|-------|
> | Original Model| 0%    | 21.81 | 6.88 | 16.00 |
> | RS-MoE        | 20%   | 18.09 | 4.47 | 13.48 |
> | RS-MoE        | 40%   | 17.56 | 4.24 | 13.19 |
> | RS-MoE        | 60%   | 15.76 | 3.98 | 12.75 |
>
> Even at a severe compression ratio of 60%, the ROUGE-L score only decreases marginally from 16.00 to 12.75. This confirms that RS-MoE successfully preserves the ability to perform complex reasoning and generate coherent text.
>
> **Action:** We have added the summarization results in Table 10 of Appendix A.7.
>
> >**W3:** Need efficiency improvement evaluation. The proposed method might reproduce too many small matrix and thus cause significant performance degradation during the generation.
>
> **A3:**
> We recognize the reviewer's concern regarding potential performance degradation due to matrix fragmentation .
>
> However, our approach avoids the inefficiencies typically associated with unstructured sparsity. In RS-MoE, the sparse components correspond to structured rows and columns which are extracted and processed as compact dense blocks. Consequently, the acceleration is achieved by replacing large weight matrices with decomposed smaller matrices and leveraging existing hardware capabilities known as dense kernels. This avoids the latency and memory irregularity often seen with sparse operations.
>
> Our empirical analysis confirms that this reduction in FLOPs translates to real-world speedups. Table 5 details the online inference efficiency:
>
> **Table 5: Online Inference Efficiency.**
> | Model | Ratio | Gate (ms) | Up (ms) | Down (ms) | Total (speedup) |
> | :--- | :---: | :---: | :---: | :---: | :---: |
> | **DeepSeekMoE-16B** | 0% | 0.69 | 0.69 | 0.73 | 2.11 |
> | | 20% | 0.65 | 0.65 | 0.69 | 1.99 (1.06x) |
> | | 40% | 0.47 | 0.47 | 0.49 | 1.43 (1.48x) |
> | | 60% | 0.36 | 0.37 | 0.41 | 1.14 (1.85x) |
> | **Mixtral-8x7B** | 0% | 13.35 | 14.09 | 16.55 | 43.99 |
> | | 20% | 10.96 | 10.88 | 11.35 | 33.19 (1.33x) |
> | | 40% | 8.09 | 8.12 | 9.75 | 25.96 (1.69x) |
> | | 60% | 5.65 | 5.81 | 6.21 | 17.67 (2.49x) |
> | **Qwen3-30B-A3B** | 0% | 0.39 | 0.40 | 0.43 | 1.22 |
> | | 20% | 0.40 | 0.40 | 0.44 | 1.24 (0.98x) |
> | | 40% | 0.34 | 0.36 | 0.37 | 1.07 (1.14x) |
> | | 60% | 0.21 | 0.21 | 0.23 | 0.65 (1.88x) |
>
> For Mixtral-8x7B, we observe a substantial acceleration of 2.49x at the 60% compression ratio. For Qwen3-30B, the speedup reaches 1.88x at 60% sparsity. This demonstrates that RS-MoE is practical and delivers significant efficiency gains on standard hardware.
>
> **Action:** We have revised the efficiency analysis in Section 4.4 and added a detailed computational cost analysis in Appendix A.5.

---

### Author Response · Authors · 2025-11-26
**Response to All Reviewers**

We sincerely thank all reviewers for their detailed and constructive feedback on our work. We are glad that reviewers acknowledged our problem identification and collaborative decomposition framework as "clear and well-motivated" (R-FPVr), "MoE-aware design" (R-YZg1), and a "practical innovation" (R-KXhS) to a significant problem.

The primary concerns raised by the reviewers centered on: (1) the specific breakdown of component contributions (e.g., ACI components and Collaborative vs. Independent strategies); (2) detailing the sensitivity and stability of our metrics; and (3) clarifications regarding practical inference latency and offline costs. Importantly, these points do not challenge the fundamental validity of our design but rather request further elucidation and supplementary empirical evidence to strengthen the paper. We have addressed each point individually through new experiments, and we believe all concerns have been fully resolved.

We respectfully request a re-evaluation of our work, taking into account these comprehensive clarifications alongside our core contributions, which remain innovative and well-substantiated. We respectfully request a re-evaluation of our submission in light of these improvements.

---

### Author Response · Authors · 2025-12-01
**Reviewer Responses and Author Rebuttal Summary**

Dear PCs, SACs, ACs, and Reviewers,

Thank you very much for your valuable contributions to our work. To assist the newly assigned AC and help reduce their workload, we provide below a summary of the key points from the reviews and the reviewer-author discussions.
***
**Strength.** Overall, we are grateful for the reviewers' constructive feedback and valuable suggestions in the initial reviews. Specifically:

- **This paper proposes a novel collaborative decomposition framework that effectively preserves expert specialization by coupling the expert weights ($W_{gate}, W_{up}, W_{down}$).**

    Four reviewers recognized this point (YZg1: Strength 1, FPVr: Strength 1, BD9Z: Strength 1, KXhS: Strength 1).

- **The proposed methodology is comprehensive and well-integrated, particularly the Anomalous Contribution Integration (ACI) and the combined sparse and low-rank strategy.**

    Four reviewers recognized this point (YZg1: Strength 2 & 3, FPVr: Strength 2, BD9Z: Strength 2, KXhS: Strength 1).

- **The experiments are extensive, solid, and demonstrate state-of-the-art performance.**

    Three reviewers explicitly highlighted this (DdvH: Strength 2, FPVr: Strength 3, YZg1: Strength 5).

- **The paper is clearly written, easy to follow, and provides sufficient details for reproducibility.**

    Three reviewers explicitly highlighted this (DdvH: Strength 1, BD9Z: Strength 3, FPVr: Strength 1).

***
**Concerns and Our Addressing.** During the discussion period, we actively addressed the reviewers' concerns by conducting lots of new experiments and adding comprehensive analyses to the revision. Specifically:

* **Concerns about Computational Cost and Efficiency.** Detailed analysis of offline compression costs and online inference acceleration. (DdvH: Weakness 3, YZg1: Weakness 2, FPVr: Weakness 4, BD9Z: Weakness 3, KXhS: Weakness 1)

    **Our Addressing.** We updated the Table 6 and added experiments in Appendix A.5 on (a) offline compression costs, (b) online inference acceleration. Please refer to our responses to reviewers DdvH, YZg1, FPVr, BD9Z, and KXhS for more details.

* **Concerns about Novelty and Methodological Contribution.** Whether RS-MoE merely applies the existing sparse+low-rank paradigm to MoE compression. (DdvH: Weakness 1, BD9Z: Weakness 1)

    **Our Addressing.** We clarified in Section 2.2 that RS MoE addresses the unique structural coupling in SwiGLU experts which independent methods ignore. Crucially, we added a new ablation study in Section 4.3 and Table 2 comparing collaborative decompostion with independent decomposition. Please refer to our responses to reviewers DdvH and BD9Z for more details.

* **Concerns about Robustness and Hyperparameters.** Sensitivity of ACI hyperparameters and the impact of outliers. (YZg1: Weakness 3, FPVr: Weakness 2, KXhS: Question 2)

    **Our Addressing.** We provided a comprehensive sensitivity analysis in Appendix A.6 and clarified that (a) our method remains stable within a robustness zone, and (b) the new Table 9 confirms the importance of activation peaks and the downstream influence. Please refer to our responses to reviewers YZg1, FPVr, and KXhS for more details.

* **Concerns about Evaluation Scope.** Including generative tasks such as summarization and checking for mode collapse. (DdvH: Weakness 2, KXhS: Question 3)

    **Our Addressing.** We added Appendix A.7 and Table 10 showing ROUGE scores on CNN DailyMail. Regarding routing consistency, we added Appendix A.8 and Figure 11. Please refer to our responses to reviewers DdvH and KXhS for more details.

* **Clarifications on Presentation and Theory.** Theoretical error bounds, the Figure 4 legend error, and notation clarity. (FPVr: Weakness 3, BD9Z: Weakness 2 & 4)

    **Our Addressing.** We addressed the challenges associated with deriving closed-form error bounds for the SwiGLU structure by providing strong empirical validation in our new Table 2. Furthermore, we corrected the swapped legend in Figure 4 and significantly clarified the notation and mathematical formulation within Section 3.4 and Section 4.4. Please refer to our responses to reviewers FPVr and BD9Z for more details.
***
Above, we have faithfully summarized all reviewer comments and our corresponding responses, hoping that this will assist the AC's work. We believe that the additional experiments and clarifications provided in the revision have effectively addressed the concerns raised. We are deeply grateful to the reviewers, AC, SAC, and PC, for their dedicated effort and excellent work. Their insightful feedback has further strengthened our paper. The authors offer their sincere respect and appreciation to all involved!

Sincerely,
Authors

---

### Meta-Review · Area_Chair_e1u3 · 2026-01-11

**Summary:**

I recommend rejection for this submission, primarily due to unresolved concerns about novelty. While the authors have conducted extensive experiments and provided thorough responses during rebuttal, it seems that RS-MoE may be more of an engineering adaptation of established sparse plus low-rank methods to MoE architectures rather than a genuine algorithmic advance. I found the distinction from prior work like LoSparse and SoLA relies heavily on empirical performance differences, without clear theoretical justification for why the proposed collaborative decomposition is fundamentally different. The absence of error analysis, which is considered essential for a compression paper, further weakens the contribution. As a result, I appreciate the authors' effort and encourage them to clarify the theoretical grounding and provide reconstruction error analysis in future revisions.

**Reviewer Concerns:**

Reviewers questioned whether the collaborative decomposition constitutes genuine novelty given prior work on sparse plus low-rank compression, with the distinction relying primarily on empirical comparisons rather than algorithmic or theoretical differentiation. The complete absence of error analysis (no reconstruction error decomposition, error propagation investigation, or connection between ACI scores and compression quality) represents a significant gap for a compression paper. Claims about preserving expert specialization lack direct validation; the routing consistency analysis measures load balancing rather than whether experts retain their specialized functions. The sensitivity analysis covers only one model-dataset configuration, providing limited guidance for practitioners, and discrepancies in Figure 4 regarding calibration sample robustness raise additional concerns about revision quality.

The authors are commended for their extensive experimental work and thorough responses during the rebuttal period, including additional ablations and analyses. However, the fundamental novelty concern (whether RS-MoE represents a genuine algorithmic advance over established sparse plus low-rank methods like LoSparse and SoLA or primarily an engineering adaptation to MoE architectures) remains insufficiently resolved, with empirical performance differences substituting for clear mechanistic or theoretical justification of the contribution.

**Reviewer Scores:**

The initial scores of 2, 4, 4, 6, and 8 reflect substantial disagreement among reviewers about the contribution's significance. While the most positive reviewer advocated for acceptance based on practical utility, the majority of reviewers remained at or below the acceptance threshold due to unresolved novelty and mechanistic understanding concerns. The overall sentiment indicates that despite improved empirical rigor through additional experiments, the paper does not meet the bar for acceptance at this venue.

---

### Decision · Program_Chairs · 2026-01-26

Reject